# Effects of a Peptide Derived from the Primary Sequence of a Kallikrein Inhibitor Isolated from *Bauhinia bauhinioides* (pep-BbKI) in an Asthma–COPD Overlap (ACO) Model

**DOI:** 10.3390/ijms241411261

**Published:** 2023-07-09

**Authors:** Luana Laura Sales da Silva, Jéssica Anastácia Silva Barbosa, Juliana Morelli Lopes Gonçalves João, Silvia Fukuzaki, Leandro do Nascimento Camargo, Tabata Maruyama dos Santos, Elaine Cristina de Campos, Arthur Silva Costa, Beatriz Mangueira Saraiva-Romanholo, Suellen Karoline Moreira Bezerra, Fernanda Tenório Quirino dos Santos Lopes, Camila Ramalho Bonturi, Maria Luiza Vilela Oliva, Edna Aparecida Leick, Renato Fraga Righetti, Iolanda de Fátima Lopes Calvo Tibério

**Affiliations:** 1Faculdade de Medicina FMUSP, Universidade de São Paulo, São Paulo 01246-903, Brazil; luana.laurasales@gmail.com (L.L.S.d.S.); jes.anastacia@gmail.com (J.A.S.B.); julianamorellift@gmail.com (J.M.L.G.J.); sfukuzaki@gmail.com (S.F.); leandro.camargo@outlook.com.br (L.d.N.C.); tabatamaruyama@gmail.com (T.M.d.S.); fisio.elaine@hotmail.com (E.C.d.C.); arthur.costa@fm.usp.br (A.S.C.); beatriz.msaraiva@fm.usp.br (B.M.S.-R.); suellen_karoline.m@hotmail.com (S.K.M.B.); fernandadtqsl@gmail.com (F.T.Q.d.S.L.); leick51@yahoo.com.br (E.A.L.); refragar@gmail.com (R.F.R.); 2Department of Medicine, University City of São Paulo, São Paulo 03071-000, Brazil; 3Departamento de Bioquímica, Universidade Federal de São Paulo (UNIFESP), São Paulo 04039-002, Brazil; camilabntr@gmail.com (C.R.B.); mlvoliva@unifesp.br (M.L.V.O.); 4Hospital Sírio-Libanês, São Paulo 01308-050, Brazil

**Keywords:** airway remodeling, ACO, Bauhinia, inflammation, oxidative stress, serine proteinase inhibitors

## Abstract

(1) There are several patients with asthma–COPD overlap (ACO). A peptide derived from the primary sequence of a kallikrein inhibitor isolated from *Bauhinia bauhinioides* (pep-BbKI) has potent anti-inflammatory and antioxidant effects. Purpose: To investigate the effects of pep-BbKI treatment in an ACO model and compare them with those of corticosteroids. (2) BALB/c mice were divided into groups: SAL (saline), OVA (ovalbumin), ELA (elastase), ACO (ovalbumin + elastase), ACO-pep-BbKI (treated with inhibitor), ACO-DX (dexamethasone treatment), ACO-DX-pep-BbKI (both treatments), and SAL-pep-BbKI (saline group treated with inhibitor). We evaluated: hyperresponsiveness to methacholine, bronchoalveolar lavage fluid (BALF), exhaled nitric oxide (eNO), IL-1β, IL-4, IL-5, IL-6, IL-10, IL-13, IL-17, IFN-γ, TNF-α, MMP-9, MMP-12, TGF-β, collagen fibers, iNOS, eNO, linear mean intercept (Lm), and NF-κB in airways (AW) and alveolar septa (AS). (3) ACO-pep-BbKI reversed ACO alterations and was similar to SAL in all mechanical parameters, Lm, neutrophils, IL-5, IL-10, IL-17, IFN-γ, TNF-α, MMP-12 (AW), collagen fibers, iNOS (AW), and eNO (*p* > 0.05). ACO-DX reversed ACO alterations and was similar to SAL in all mechanical parameters, Lm, total cells and differentials, IL-1β(AS), IL-5 (AS), IL-6 (AS), IL-10 (AS), IL-13 (AS), IFN-γ, MMP-12 (AS), TGF-β (AS), collagen fibers (AW), iNOS, and eNO (*p* > 0.05). SAL was similar to SAL-pep-BbKI for all comparisons (*p* > 0.05). (4) Pep-BbKI was similar to dexamethasone in reducing the majority of alterations of this ACO model.

## 1. Introduction

Asthma and COPD are heterogeneous conditions characterized by airway obstruction. They have been widely studied; however, in recent years, a variant of these diseases has been noticed, the characteristics of which overlap with those of both conditions, with similar symptoms to those found in COPD but a significant response to bronchodilators. Each one may include different inflammatory patterns and underlying mechanisms, but some of them can be common to both. Thus, this condition has been called asthma–COPD overlap (ACO) [1,2,3].

There are three common clinical features of obstructive pulmonary diseases: airway inflammation, airway obstruction, and bronchial hyperresponsiveness. In asthma, chronic inflammation is mainly characterized by an eosinophilic infiltrate and CD4+ cells [4], and, in COPD, the predominant inflammation is caused by neutrophils and CD8+ cells [5]. However, there are reports in the literature that asthmatics are non-eosinophilic and show important resistance to corticosteroids [6]. Patients with asthma who smoke have an increased number of neutrophils in their airways, such as those with COPD. Smoking induces neutrophilic inflammation, which promotes resistance to corticosteroids, and patients with COPD that use corticosteroids have an increased risk of pneumonia [7,8,9]. However, eosinophilic inflammation is commonly observed in patients with asthma and is correlated with greater reversibility of airflow obstruction when corticosteroids are administered [10]. GINA (2022) [3] suggests that asthma–COPD overlap is a descriptor for patients who comprise a heterogeneous group, which does not mean a single disease entity. Based on these similarities and differences in the definition and pathophysiology of ACO, interest in biomarkers to describe ACO has increased, and further scientific investigation is necessary.

Protease inhibitors, which can originate from plants or animals, can produce complexes with enzymes and then be able to inhibit catalytic activity [11]. Protease inhibitors are classified into families according to their primary molecular structures. They can inhibit proteolytic enzymes or increase the levels of endogenous antiproteases and may contribute to the prevention of disease progression [12,13,14].

*Bauhinia* is a plant genus belonging to the subfamily *Caesalpinioideae* which can be found in tropical regions. Protease inhibitors that have been isolated from this genus, particularly from the seeds of the species *Bauhinia bauhinioides*, which have kallikrein inhibitors, are called *Bauhinia bauhinioides* kallikrein inhibitors (BbKIs) [11,15]. BbKI is a Kunitz-type inhibitor with a molecular mass of around 18 kDa and an apparent constant inhibition (Kiap) of 2.4 nM for kallikrein and 33 nM for plasmin [16]. Kallikreins are glycoproteins present in glandular cells, neutrophils, and biological cell fluids [14]. BbKI inhibits serine proteases, which are inhibitors of trypsin, chymotrypsin, and plasmin, and is an important inhibitor of human plasma kallikrein (PKa), which is important for the release of kinins involved in inflammation [13]. Tissue kallikreins are directly involved in tumor progression through increased expression and dysregulation of proteolysis [17]. Inhibition of PKa by BbKI causes a reduction in bradykinin release by decreasing the availability of PKa [18].

Proteases are no longer considered primarily to be protein-degrading enzymes and have become important signaling molecules involved in numerous vital processes [19]. Therefore, different protease inhibitors of plant origin have been studied in asthma and COPD models [20,21,22,23,24,25,26].

The present study aimed to investigate whether peptide BbKI contributes to controlling the progression of the asthma–COPD overlap model; to study the lung mechanics of hyperresponsiveness to methacholine, inflammatory response, extracellular matrix remodeling, and oxidative stress responses; and to compare the treatment with corticosteroid treatment.

## 2. Results

The SAL and SAL-pep-BbKI control groups were not significantly different (*p* > 0.05), as shown in the Appendix A. Therefore, only the SAL group is shown in the graphs to facilitate the visualization of the results.

### 2.1. Hyperresponsiveness to Methacholine

Figure 1 shows the values for the evaluation of hyperresponsiveness to methacholine, and the results are expressed as the maximum percentage increases after the tests with methacholine in the experimental groups. There was an increase in respiratory system elastance (%Ers) in the OVA group (203.4 ± 33.7%) compared to that in the SAL (77.3 ± 7.5%) group (*p* < 0.05). There was a reduction in %Ers in the ELA (32.4 ± 3.1%) and ACO (36.4 ± 4.8%) groups compared to that in the SAL and OVA groups (*p* < 0.05). The ACO-pep-BbKI (110.3 ± 28.8%) and ACO-DX (126.1 ± 24.3%) treatment groups reversed the alterations in %Ers because they were different from ACO (*p* < 0.05) and similar to the SAL group (*p* > 0.05).

The respiratory system resistance (%Rrs) increased in the OVA (351.6 ± 47.0%) and ACO (625.1 ± 81.0%) groups compared to that in the SAL (113.7 ± 14.8%) group (*p* < 0.05). We also observed an increase in %Rrs in the ACO group compared to that in the OVA and ELA (188.9 ± 63.3%) groups (*p* < 0.05). In addition, there was a full reversal in the alterations of %Rrs in the ACO-pep-BbKI (260.7 ± 31.5%), ACO-DX (227.3 ± 70.4%), and ACO-DX-pep-BbKI (229.3 ± 40, 0%) groups because all of them were different from ACO (*p* < 0.05) and similar to the SAL group (*p* > 0.05).

The percentage of airway resistance (%Raw) was higher in the OVA (379.8 ± 62.7%), ELA (240.3 ± 38.2%), and ACO (490.2 ± 37.5%) groups than in the SAL group (94.4 ± 16.2%) (*p* < 0.05). Raw values in the ACO group were superior to those in the ELA group (*p* < 0.05). There was a reversal of alterations in Raw values in the ACO-pep-BbKI (167.4 ± 39.7%), ACO-DX (133.0 ± 59.3%), and ACO-DX-pep-BbKI (213.8 ± 57.7%) groups because they were different from ACO (*p* < 0.05) and equal to the SAL (*p* > 0.05) group.

The evaluation of lung tissue elastance (%Htis) showed that the ELA (19.5 ± 5.4%) and ACO (24.2 ± 5.0%) groups were similar to the SAL (39.0 ± 5.6%) group (*p* > 0.05). The OVA (72.9 ± 19.0%) group was superior to the SAL group (*p* < 0.05). The ELA and ACO groups showed a decrease in %Htis compared to the OVA group (*p* < 0.05). None of the treatment groups differed from the SAL or ACO groups (*p* > 0.05).

Lung tissue damping (%Gtis) increased in the OVA (114.7 ± 12.7%) and ACO (187.6 ± 21.6%) groups compared to the SAL group (54.4 ± 9.9%) (*p* < 0.05). We also observed an increase in the %Gtis of ACO compared to the OVA and ELA (61.5 ± 8.6%) groups (*p* < 0.05). In addition, there was a complete reversal in alterations in the ACO-pep-BbKI (47.8 ± 15.4%), ACO-DX (56.2 ± 17.2%), and ACO-DX-pep-BbKI (87.7 ± 18, 1%) groups because they were different from the ACO group (*p* < 0.05) and equal to the SAL group (*p* > 0.05).

### 2.2. Bronchoalveolar Lavage Fluid

The total and differential inflammatory cell counts are shown in Figure 2. The number of inflammatory cells increased in the OVA (13.6 ± 2.3 × 10^4^ cells/mL), ELA (12.6 ± 1.4 × 10^4^ cells/mL), and ACO (22.2 ± 3.7 × 10^4^ cells/mL) groups compared to the SAL group (0.8 ± 0.1 × 10^4^ cells/mL) (*p* < 0.05). There was an increase in the number of total cells in the ACO group when compared to the OVA and ELA groups (*p* < 0.05). There was attenuation in the number of total cells in the ACO-pep-BbKI (7.1 ± 1.5 × 10^4^ cells/mL) and ACO-DX-pep-BbKI (4.9 ± 0.8 × 10^4^ cells/mL) treatment groups compared to the ACO (*p* < 0.05) group, since they were different from the SAL group (*p* < 0.05). The ACO-DX (2.5 ± 0.2 × 10^4^ cells/mL) group showed a reversal of the alterations because it was different from the ACO (*p* < 0.05) and similar to the SAL (*p* > 0.05) group.

Macrophage counts increased in the OVA (2.4 ± 0.4 × 10^4^ cells/mL), ELA (6.3 ± 0.7 × 10^4^ cells/mL), and ACO (7.5 ± 1.4 × 10^4^ cells/mL) groups compared to those in the SAL (0.2 ± 0.03 × 10^4^ cells/mL) (*p* < 0.05). The macrophages in the ACO group were higher than in the OVA group (*p* < 0.05). The ACO-pep-BbKI (1.8 ± 0.4 × 10^4^ cells/mL) and ACO-DX-pep-BbKI (1.4 ± 0.2 × 10^4^ cells/mL) treatment groups attenuated macrophage counts compared to the ACO group (*p* < 0.05) because they were different from the SAL group (*p* < 0.05). The ACO-DX group (0.9 ± 0.2 × 10^4^ cells/mL) completely reversed the alterations because it was different from the ACO (*p* < 0.05) and equal to the SAL group (*p* > 0.05).

Neutrophil counts increased in the OVA (2.3 ± 0.4 × 10^4^ cells/mL), ELA (3.1 ± 0.5 × 10^4^ cells/mL), and ACO (4.3 ± 0.5 × 10^4^ cells/mL) groups compared to the SAL group (0.1 ± 0.02 × 10^4^ cells/mL) (*p* < 0.05). The ACO group had higher neutrophil counts than the OVA group (*p* < 0.05). In the ACO-pep-BbKI (1.0 ± 0.3 × 10^4^ cells/mL) and ACO-DX treatment groups (0.5 ± 0.1 × 10^4^ cells/mL), we observed a reverse in the neutrophil counts compared to the ACO group (*p* < 0.05) because they were similar to the SAL group (*p* > 0.05). ACO-DX-pep-BbKI (1.3 ± 0.4 × 10^4^ cells/mL) attenuated the numbers because it was different from the ACO and SAL groups (*p* < 0.05).

We observed an increase in the eosinophil counts in the OVA (6.6 ± 1.2 × 10^4^ cells/mL) and ACO (9.1 ± 1.4 × 10^4^ cells/mL) groups compared to the SAL group (0.2 ± 0.05 × 10^4^ cells/mL) (*p* < 0.05). The ACO group was superior to the ELA group (1.8 ± 0.2 × 10^4^ cells/mL) (*p* < 0.05). There was an attenuation in the number of eosinophils in the ACO-pep-BbKI group (2.8 ± 0.7 × 10^4^ cells/mL) compared to that in the ACO group (*p* < 0.05) because it was different from the SAL group (*p* < 0.05). ACO-DX (0.7 ± 0.1 × 10^4^ cells/mL) and ACO-DX-pep-BbKI (1.4 ± 0.2 × 10^4^ cells/mL) completely reversed the numbers because they were different from ACO (*p* < 0.05) and were similar to SAL (*p* > 0.05). The ACO-DX (0.7 ± 0.1 × 10^4^ cells/mL) group had lower eosinophil counts than the ACO-pep-BbKI group (*p* < 0.05).

There was an increase in lymphocyte counts in the OVA (1.2 ± 0.2 × 10^4^ cells/mL), ELA (1.5 ± 0.3 × 10^4^ cells/mL), and ACO (2.1 ± 0.4 × 10^4^ cells/mL) groups compared to those in the SAL group (0.2 ± 0.05 × 10^4^ cells/mL) (*p* < 0.05). Lymphocyte counts were reversed in the ACO-DX (0.3 ± 0.1 × 10^4^ cells/mL) and ACO-DX-pep-BbKI groups (0.7 ± 0.2 × 10^4^ cells/mL) when compared to the ACO group (*p* < 0.05) and were equal to the SAL group (*p* > 0.05). And there was no difference between the ACO-pep-BbKI (1.1 ± 0.2 × 10^4^ cells/mL) and ACO groups.

### 2.3. Mean Linear Intercept (Lm)

Figure 3 shows the Lm values for all experimental groups. There was an increase in Lm in the ELA (42.7 ± 1.9 μm) and ACO (59.3 ± 1.9 μm) groups compared to the SAL group (27.2 ± 2.8 μm) (*p* < 0.05). The ACO group was different from the OVA (29.1 ± 1.8 μm) and ELA groups (*p* < 0.05). In the ACO-pep-BbKI (21.4 ± 0.8 μm) and ACO-DX (29.3 ± 1.3 μm) groups, we observed a complete reversal of alterations in Lm compared with the ACO group (*p* < 0.05), as they were similar to the SAL group (*p* > 0.05). In the ACO-DX-pep-BbKI group (20.5 ± 0.8 μm), the alterations were attenuated compared to the ACO group (*p* < 0.05) and were different from those of the SAL group (*p* < 0.05). The groups treated with the protease inhibitors (ACO-pep-BbKI and ACO-DX-pep-BbKI) showed a decrease in Lm compared to the group treated with dexamethasone only (ACO-DX) (*p* < 0.05).

### 2.4. Lung Inflammation

The absolute values of inflammatory markers for the airways and alveolar septa in all experimental groups are shown in Table 1.

In the airways and alveolar septa, there was an increase in IL-1β-positive cells in the OVA and ACO groups compared to the control SAL group (*p* < 0.05). The ACO group showed a higher IL-1β count than the OVA and ELA groups (*p* < 0.05). In the airways, the ACO-pep-BbKI, ACO-DX, and ACO-DX-pep-BbKI groups showed attenuation of IL-1β-positive cells when compared to the ACO (*p* < 0.05) group, and they were different from the SAL (*p* < 0.05) group. In the alveolar septa, ACO-DX reversed the alterations compared to the ACO group (*p* < 0.05) once it became similar to the SAL group (*p* > 0.05).

The number of IL-4 positive cells in the airways and alveolar septa increased in all groups compared to the control SAL group (*p* < 0.05) and decreased in the airways compared to the SAL and ELA groups (*p* > 0.05). The ACO group showed an increase in IL-4 counts compared to the OVA and ELA groups in the alveolar septa analysis and was only different from the ELA group (*p* < 0.05) with respect to airways. However, the treatment groups did not show decreases in the counts for IL-4 compared to the ACO group (*p* > 0.05).

IL-5 positive cells increased in the ACO, OVA, and ELA groups compared to the SAL group (*p* < 0.05) in the airways and alveolar septa. ACO showed superior cell counts compared to OVA and ELA (*p* < 0.05). In alveolar septa, all treatment groups, ACO-pep-BbKI, ACO-DX, and ACO-DX-pep-BbKI, reversed IL-5 counts compared to the ACO group (*p* < 0.05), and were similar to the SAL group (*p* > 0.05). In the airways, a difference was exhibited by the ACO-DX group, in which the alterations were attenuated compared to the ACO group, and it was different from the SAL group (*p* < 0.05). A comparison of the treatments groups, only with respect to the airways, showed that IL-5-positive cells decreased in the ACO-pep-BbKI and ACO-DX-pep-BbKI groups compared to the group treated with corticosteroids only (ACO-DX) (*p* < 0.05).

In the airways and alveolar septa, there was an increase in IL-6-positive cells in the OVA, ELA, and ACO groups compared with the control group (SAL) (*p* < 0.05). Only in the alveolar septa did ACO increase in these cells compared to OVA and ELA (*p* < 0.05). In the airways, all treatment groups (ACO-pep-BbKI, ACO-DX, and ACO-DX-pep-BbKI) showed attenuation of IL-6 compared to the ACO group (*p* < 0.05), as they were different from SAL (*p* < 0.05). In alveolar septa only, the ACO-DX and ACO-DX-pep-BbKI groups reversed the IL-6 alterations compared to the ACO (*p* < 0.05) group because they were similar to the SAL (*p* > 0.05). These groups that received corticosteroids (ACO-DX and ACO-DX-pep-BbKI) showed a decrease in IL-6-positive cells compared to the group that received only the inhibitor (the ACO-pep-BbKI group) (*p* < 0.05).

IL-10-positive cells in the airways and alveolar septa showed an increase in the OVA, ELA, and ACO groups compared to the SAL group (*p* < 0.05), and the ACO group showed a higher count of IL-10 positive cells than the OVA group (*p* < 0.05). In the evaluation of airways, the groups treated with the inhibitor (ACO-pep-BbKI and ACO-DX-pep-BbKI) showed reversed alterations of IL-10-positive cells when compared to the ACO group (*p* < 0.05), and they were similar to the SAL group (*p* > 0.05). In addition, they were different from the ACO-DX group (*p* < 0.05). In the alveolar septa, all the treatment groups showed reversal of the alterations in IL-10-positive cells compared to the ACO group (*p* < 0.05), and they were similar to the SAL group (*p* > 0.05). However, the ACO-pep-BbKI group showed a lower count of these positive cells compared to the groups treated with dexamethasone (ACO-DX and ACO-DX-pep-BbKI) (*p* < 0.05).

The count of IL-13-positive cells in the airways and alveolar septa was higher in the OVA, ELA, and ACO groups than in the control group (SAL) (*p* < 0.05). The alveolar septa of the ACO group increased compared to the OVA and ELA groups, and in the airways, they were different from the ELA group only (*p* < 0.05). With respect to the airways, the ACO-pep-BbKI, ACO-DX, and ACO-DX-pep-BbKI groups showed attenuation of the IL-13-positive cells compared to the ACO group (*p* < 0.05), and they were different from the SAL group (*p* < 0.05). In the alveolar septa analysis, ACO-DX reversed the alteration compared to the ACO group (*p* < 0.05), and it was similar to the SAL group (*p* > 0.05).

Furthermore, we observed an increase in IL-17-positive cells in the OVA, ELA, and ACO groups in the airways and alveolar septa compared to the SAL group (*p* < 0.05). The ACO group showed higher positive cells count for IL-17 than the OVA group (*p* < 0.05). The ACO-DX and ACO-DX-pep-BbKI treatment groups had attenuated IL-17-positive cells in comparison to the ACO group (*p* < 0.05), and they were different from the SAL group (*p* < 0.05). The ACO-pep-BbKI group reversed the alteration compared to the ACO group (*p* < 0.05), and was equal to the SAL group (*p* > 0.05). The alveolar septa analysis of the treated groups showed that the groups treated with BbKI (ACO-pep-BbKI and ACO-DX-pep-BbKI) showed a decrease in these cells compared to the group treated with dexamethasone only (ACO-DX) (*p* < 0.05).

In the airways and alveolar septa, IFN-γ-positive cells increased in the ELA, OVA, and ACO groups compared to the SAL group (*p* < 0.05). In the airways, the ACO group showed an increased IFN-γ count compared to the OVA and ELA groups (*p* < 0.05). The treatment groups ACO-pep-BbKI and ACO-DX showed complete reversal of the number of IFN-γ-positive cells compared to the ACO group (*p* < 0.05), and they were equal to the SAL group (*p* > 0.05). Only in the airways, IFN-γ decreased in the ACO-DX group compared to the other two treatment groups (ACO-pep-BbKI and ACO-DX-pep-BbKI) (*p* < 0.05).

In the alveolar septa and airways, there was an increase in TNF-𝛼-positive cells in the OVA, ELA, and ACO groups compared with the control group (SAL) (*p* < 0.05), and only in the alveolar septa was the ACO group different from the OVA group (*p* < 0.05). In the alveolar septa, the ACO-pep-BbKI, ACO-DX, and ACO-DX-pep-BbKI groups had attenuated TNF-𝛼-positive cells compared to ACO (*p* < 0.05), and they were different from the SAL group (*p* < 0.05). We also observed that positive cells decreased in the ACO-pep-BbKI group compared to the ACO-DX and ACO-DX-pep-BbKI groups (*p* < 0.05).

### 2.5. Extracellular Matrix Remodeling

The values of the remodeling markers in all experimental groups are shown in Table 2 for the airways and alveolar septa.

When evaluating MMP-9-positive cells in the airways and alveolar septa, we observed an increase in these cells in the OVA, ELA, and ACO groups compared with the SAL group (*p* < 0.05), which was also observed in the ACO group compared to the OVA and ELA groups (*p* < 0.05). The ACO-pep-BbKI, ACO-DX, and ACO-DX-pep-BbKI treatment groups had attenuated numbers of MMP-9-positive cells compared to the ACO group (*p* < 0.05), and they were different from the SAL group (*p* > 0.05). In addition, the group treated with pep-BbKI and dexamethasone showed a greater reduction than that treated with dexamethasone alone (*p* < 0.05).

In the airways and alveolar septa, the MMP-12-positive cells increased in the ACO group compared to the OVA, ELA, and SAL groups (*p* < 0.05). In the airways, all the treatment groups, ACO-pep-BbKI, ACO-DX, and ACO-DX-pep-BbKI, showed reversed numbers of MMP-12-positive cells compared to the ACO group (*p* < 0.05), and they were similar to the SAL group (*p* > 0.05). In the alveolar septa, the ACO-pep-BbKI and ACO-DX groups showed attenuated numbers compared to the ACO group (*p* < 0.05), and they were different from the SAL group (*p* < 0.05). In the airways, the groups that received corticosteroids, ACO-DX and ACO-DX-pep-BbKI, showed decreased MMP-12-positive cells compared to the group treated only with pep-BbKI (*p* < 0.05). In the alveolar septa, the groups treated with both dexamethasone and pep-BbKI showed a reduction in MMP-12-positive cells compared to ACO-pep-BbKI and ACO-DX separately (*p* < 0.05).

The evaluation of TGF-β-positive cells in the airways showed an increase in these cells in the OVA and ACO groups compared to the SAL group, and in the ACO group increased compared to the ELA (*p* < 0.05). The ACO-pep-BbKI and ACO-DX groups showed attenuated TGF-β compared to the ACO group (*p* < 0.05) because they were different from SAL (*p* < 0.05). TGF-β decreased in the ACO-DX-pep-BbKI than in the group treated only with the inhibitor (ACO-pep-BbKI) (*p* < 0.05). In alveolar septa, the ACO showed an increase in TGF-β when compared to the SAL, OVA and ELA groups. The ACO-pep-BbKI and ACO-DX-pep-BbKI groups attenuated the values of TGF-β-positive cells compared to the ACO group (*p* < 0.05), and they were different from the SAL (*p* < 0.05). The ACO-DX group reversed the alterations because it was different from the ACO group (*p* < 0.05) and similar to the SAL (*p* > 0.05), as well as showed a reduction in TGF-β-positive cells compared to the ACO-pep-BbKI and ACO-DX-pep-BbKI groups (*p* < 0.05).

In the airways and alveolar septa, there was an increase in the volume fraction of collagen fibers in the OVA and ACO groups compared with that in the SAL group (*p* < 0.05). On the contrary, in the ELA group, there was a reduction in the volume fraction of collagen fibers compared to that in the ACO the group (*p* < 0.05). In the airways, all the treatment groups reversed the alterations in the volume fraction of collagen compared to the ACO group (*p* < 0.05), because they were similar to the SAL group (*p* > 0.05). In the alveolar septa, the ACO-pep-BbKI and ACO-DX-pep-BbKI groups reversed these alterations compared to the ACO group (*p* < 0.05), because they were equal to the SAL (*p* > 0.05). However, the ACO-DX was attenuated compared to the ACO group because it differed from the SAL group (*p* < 0.05). We also observed that the groups that contained the protease inhibitors (ACO-pep-BbKI and ACO-DX-pep-BbKI) showed a reduction in the volume of collagen fibers compared to the group treated only with dexamethasone (ACO-DX) (*p* < 0.05).

### 2.6. Oxidative Stress Response

The absolute values of the oxidative stress markers for all experimental groups are shown in Figure 4. In the airways, there was an increase in this number of iNOS-positive cells in the OVA (9.9 ± 1.3 cells/10^4^ µm^2^), ELA (11.8 ± 0.9 cells/10^4^ µm^2^) and ACO (8.6 ± 0.5 cells/10^4^ µm^2^) groups compared that in the SAL group (3.3 ± 0.5 cells/10^4^ µm^2^) (*p* < 0.05). We observed an increase in the number of iNOS-positive cells in the ELA group compared with the ACO group (*p* < 0.05). All the treatment groups, ACO-pep-BbKI (4.5 ± 0.5 cells/10^4^ µm^2^)*,* ACO-DX (3.3 ± 0.3 cells/10^4^ µm^2^), and ACO-DX-pep-BbKI (2.5 ± 0.3 cells/10^4^ µm^2^)*,* reversed the alterations of iNOS-positive cells compared with the ACO group (*p* < 0.05), because they were similar to the SAL group (*p* > 0.05). The group that received treatment with corticosteroids and the pep-BbKI group showed a decrease in iNOS-positive cells compared to the group treated with BbKI only (*p* < 0.05).

We observed similar results in the alveolar septa; there was an increase in iNOS-positive cells in the OVA (8.6 ± 0.9 cells/10^4^ µm^2^), ELA (9.2 ± 0.8 cells/10^4^ µm^2^), and ACO (9.4 ± 0.5 cells/10^4^ µm^2^) groups compared to that in the SAL group (2.6 ± 0.4 cells/10^4^ µm^2^) (*p* < 0.05). In the treatment groups, ACO-pep-BbKI (4.3 ± 0.5 cells/10^4^ µm^2^) attenuated the alterations compared to ACO (*p* < 0.05), because it was different from the SAL group (*p* < 0.05). ACO-DX (2.8 ± 0.3 cells/10^4^ µm^2^) and ACO-DX-pep-BbKI (3.1 ± 0.3 cells/10^4^ µm^2^) reversed the alterations of iNOS-positive cells compared with the ACO group (*p* < 0.05), as they were similar to the SAL group (*p* > 0.05).

### 2.7. Exhaled Nitric Oxide (eNO)

Figure 4 shows the exhaled nitric oxide (eNO) values in all experimental groups. There was an increase in eNO in the OVA (32.4 ± 4.6 ppb), ELA (26.8 ± 3.8 ppb) and ACO (41.0 ± 5.1 ppb) groups compared to that in the SAL group (12.3 ± 2.1 ppb) (*p* < 0.05). However, all the treatment groups, ACO-pep-BbKI (17.1 ± 5.7 ppb)*,* ACO-DX (21.2 ± 3.9 ppb), and ACO-DX-pep-BbKI (16.1 ± 2.7 ppb), reversed the alterations of eNO compared with the ACO group (*p* < 0.05), because they were similar to SAL (*p* > 0.05).

### 2.8. Signaling Pathway

In the airways and alveolar septa, there was an increase in NF-κB-positive cells in the OVA (6.7 ± 0.6 cells/10^4^ µm^2^—airways; 6.2 ± 0.4 cells/10^4^ µm^2^—alveolar septa), ELA (8.6 ± 0.7 cells/10^4^ µm^2^—airways; 9.0 ± 0.5 cells/10^4^ µm^2^—alveolar septa) and ACO (6.5 ± 0.4 cells/10^4^ µm^2^—airways; 5.9 ± 0.4 cells/10^4^ µm^2^—alveolar septa) groups compared to the SAL group (0.4 ± 0.1 cells/10^4^ µm^2^—airways; 0.2 ± 0.1 cells/10^4^ µm^2^—alveolar septa) (*p* < 0.05). The ELA group had an increase in the number of NF-κB-positive cells compared to the ACO group (*p* < 0.05). All the treatment groups, ACO-pep-BbKI (4.1 ± 0.5 cells/10^4^ µm^2^—airways; 2.8 ± 0.4 cells/10^4^ µm^2^—alveolar septa), ACO-DX (2.3 ± 0.2 cells/10^4^ µm^2^—airways; 2.0 ± 0.3 cells/10^4^ µm^2^—alveolar septa) and ACO-DX-pep-BbKI (2.3 ± 0.3 cells/10^4^ µm^2^—airways; 1.6 ± 0.3 cells/10^4^ µm^2^—alveolar septa), attenuated the number of NF-κB-positive cells compared to ACO group (*p* < 0.05) because they were different from the SAL group (*p* < 0.05).

In the evaluation of the airways, the group that received treatment within corticosteroids and even with the pep-BbKI together showed a decrease in the numbers of NF-κB-positive cells compared to that treated with BbKI only (*p* < 0.05). In the alveolar septa, the ACO-DX-pep-BbKI group showed a decrease in NF-κB-positive cells compared to ACO-pep-BbKI (*p* < 0.05). All these results are shown in Figure 4.

### 2.9. Qualitative Analysis

Figure 5 and Figure 6 show representative photomicrographs of the inflammatory markers, remodeling markers, oxidative stress and signaling pathways reflected by positive cells stained for IL-5, MMP-12, iNOS, and NF-κB, in the airways and alveolar septa, respectively.

## 3. Discussion

In our study, we showed that pep-BbKI and dexamethasone were able to attenuated the hyperresponsiveness to methacholine, some inflammatory cells, reduction of extracellular cellular matrix remodeling, oxidative stress markers, and NF-κB-positive cells in the airways and alveolar septa in a model of allergic pulmonary inflammation and elastase-induced emphysema overlap (ACO model).

New therapies have been studied using models of emphysema or asthma, but experimental models for asthma and emphysema overlap still lacking in the literature. In this study, our model of asthma-COPD overlap was modified from that of Ikeda et al. [27]. Here, we demonstrated that the alveolar enlargement of lung tissue, inflammation, and remodeling in mice induced by ovalbumin and elastase constitute an interesting research model for possible therapeutic strategies for asthma-COPD overlap.

Previous studies have shown that plant-derived protease inhibitors have anti-inflammatory and antioxidant properties and have improved pulmonary inflammation, oxidative stress, and remodeling in experimental models of elastase-induced emphysema and allergic pulmonary inflammation [20,21,22,23].

Some of our findings align with a study by Ikeda et al. [27], which used a combined model of ovalbumin and elastase in mice and observed increased airway hyperresponsiveness and static compliance. However, they did not evaluate other parameters such as Raw, Htis, and Gtis, which describe the viscoelastic properties of the respiratory system and lung tissue resistance. In our study, we demonstrated that treatment with pep-BbKI and dexamethasone reduced hyperresponsiveness to methacholine as assessed by Rrs, Raw, and Gtis, indicating bronchodilatory effects of pep-BbKI and decreased lung resistance. We also observed that pep-BbKI and dexamethasone reversed the alterations in Ers observed in the ACO group, suggesting attenuation of parenchymal destruction and improvement in recoil elasticity.

Some studies of elastase-induced emphysema corroborate our findings. Martins-Oliveira et al. [23], found a reduction in the Raw and Rrs with BbKI treatment (it was used asprotein, and not the peptide), while other studies using protease inhibitors, such as *Crataeva tapia Bark Lectin* (CrataBL), *Bauhinia bauhinioides* cruzipain proteinase inhibitor (BbCI), and *Enterolobium contortisiliquum* (EcTI), found a good response in the reduction of Raw_,_ but no differences were seen in Rrs and Gtis analysis [20,24,25]. Studies in the asthma model also found results similar to ours, treatment with protease inhibitors such as CrataBL, EcTI, and *Boophilus microplus trypsin inhibitor* (rBmTI-A), they showed a reduction in Rrs and Ers [21,22,26].

However, the authors have observed different effects on lung mechanics in elastase-treated models when assessing the Ers and Htis. Previous studies on animals receiving intratracheal elastase showed an increase in Ers and Htis, and they found a decrease in these parameters after treatment with serine proteases inhibitors such as BbKI, CrataBL, BbCI, and EcTI [20,23,24,25]. The different results in our study in the reduction of Ers and Htis in the ELA and ACO groups, may be due to the time between the day of administration of intratracheal elastase and the lung mechanics. In the present study, it was seven days, while in the previously cited studies, it was 28 days. The time for tissue remodeling was shorter in our study, with greater tissue injury, less collagen fibers content, and a decrease in Ers and Htis in the emphysema and overlap groups.

Ito et al. [28,29] conducted studies on mice receiving elastase through nebulization and direct instillation and observed a decrease in respiratory system elastance. Despite a significant increase in total collagen content in the lung, Ers decreased, indicating abnormal collagen function. They suggested that abnormal collagen remodeling plays a major role in pulmonary function and mechanical forces associated with emphysema.

Despite these differing results, Scuri et al. [30] showed that elastase increases the production of bradykinin by activating tissue kallikreins, causing bronchoconstriction by increasing lung resistance and elastance. Furthermore, these authors showed that treatment with a histamine antagonist did not reverse this response, but it was blocked by antagonists of the bradykinin B2 receptor, suggesting the participation of the kallikrein–kinin system in the process. In the present study, we showed that pep-BbKI ameliorated lung function, equaling the elastance of the ACO-pep-BbKI group to that of control group (SAL). BbKI, in turn, blocks the activity of human and rat plasma kallikreins, trypsin, chymotrypsin, and plasmin, and it is a unique inhibitor, so far, isolated from plants that inhibits the tissue kallikrein activity [13,17].

Regarding the inflammatory process assessed by bronchoalveolar lavage fluid, we found a notably decrease in the number of total cells, macrophages, neutrophils and eosinophils in the ACO-pep-BbKI and ACO-DX groups compared to the ACO group, indicating an important role of the protease inhibitor (pep-BbKI) as well as dexamethasone in the modulation of anti-inflammatory processes. However, lymphocyte count decreased with dexamethasone treatment but not with pep-BbKI alone, and the comparing treatments showed that dexamethasone had a better decrease in eosinophils counts than pep-BbKI, an important effect of corticosteroids in improving the inflammatory response.

These findings align with a study by Martins-Oliveira et al. [23], where treatment with BbKI (as a protein structure) reduced the counts of total cells, neutrophils, eosinophils, and macrophages in bronchoalveolar lavage fluid (BALF) in an emphysema model. However, they also found a decrease in lymphocytes, but their analysis focused solely on the emphysema model, while our study evaluated the overlap of asthma-COPD. Another study involving CrataBL, a trypsin inhibitor, demonstrated reduced inflammatory cell response. In an asthma model, the authors observed a decrease in total cells, macrophages, and lymphocytes, while in an emphysema model, they found a decrease in neutrophils, lymphocytes, and eosinophils [21,25]. Almeida-Reis et al. [20] showed a reduction in total cells and neutrophils using BbCI, a Kunitz family inhibitor, as a treatment for elastase-induced emphysema. Similar results were reported by Neuhof et al. [31] in a lung edema model caused by neutrophil activation, where the BbCI elastase inhibitor reduced edema formation.

Florencio et al. [22] and Lourenço et al. [32] studied another serine protease inhibitor, the rBmTI-A. In an experimental asthma model, the first author observed a reduction in the concentration of eosinophils, but there was no reduction in lymphocytes. The second author studied this peptidase inhibitor in an experimental emphysema model and found a reduction in the counts of macrophages but no reduction in lymphocyte counts either.

We found an effective decrease in alveolar enlargement in all treatment groups of our study; the ACO-pep-BbKI and ACO-DX-pep-BbKI groups showed better attenuation of the linear mean intercept than the group treated only with dexamethasone. This result suggests that pep-BbKI decreases the parenchymal lesions. Lm is a measurement of the average space between opposing alveolar walls, and emphysema is characterized by alveolar wall destruction [33]. Several studies on protease inhibitors corroborate our findings on Lm improvement [20,23,25].

Regarding the evaluation of inflammatory markers in the airways, we observed that the ACO group, compared to the asthma model (OVA), was higher in cells for IL-1β, IL-10, IL-17, and IFN-γ, with the exception of IL-5 positive cells that were superior in the OVA group. Compared to the emphysema model (ELA), the ACO group was superior in expressing of IL-1β, IL-4, IL-5, IL-6, IL-13, and IFN-γ. When evaluating the alveolar septa, the ACO group showed superior counts for IL-1β, IL-5, IL-6, IL-10, IL-13, IL-17, and TNF-α positive cells, when compared to the OVA group, and compared to the ELA group, IL-1β, IL-4, IL-5, IL-6 and IL-13 were higher in ACO group. Similar findings by Ikeda et al. [27] corroborated our findings, they showed that the group that received OVA and elastase in combination displayed a significant increase in the AHR, leukotriene levels, and CD4^+^ and CD8^+^ T cell counts in the BALF compared to animals that received airway challenge asthma or elastase-induced emphysema alone. These results suggested that the lungs were more inflamed in the overlap phenotype than in the asthma and emphysema models.

In the present study, there was an increase in the inflammatory markers involved, such as IL-1β, IL-4, IL-5, IL-6, IL-10, IL-13, IL-17, IFN-γ, and TNF-α, in the experimental model of asthma-COPD overlap. These cytokines are directly linked to the inflammatory processes in asthma and COPD as known. We obtained a good response to treatment with pep-BbKI, dexamethasone, and the combination of both when we evaluated all these cytokines in comparison with the ACO group.

We observed that in the airways, the attenuation of IL-1β and IFN-γ positive cells was enhanced in the group that was treated with dexamethasone when compared to pep-BbKI alone, but in the analysis, the treatments were equally efficient in the alveolar septa. Raundhal et al. [34] suggested that high IFN-γ levels promote airway hyperresponsiveness through the suppression of secretory leukocyte protease inhibitor expression in bronchial epithelial cells and that IFN-γ mediates the immune response can differentiate severe asthma from mild-moderate asthma in both humans and mice. Corroborating with these results, our study showed a reduction of IFN-γ positive cells and a decrease in airway hyperresponsiveness. However, we observed that IFN-γ positive cells were higher in the ACO group than the OVA and ELA groups separated, in the airways. These findings were also observed in animal models of asthma [26,35,36]. IL-1β has recently been implicated in severe asthma and is associated with exacerbations of chronic obstructive pulmonary disease (COPD); however, the underlying mechanisms remain unclear. Mahmutovic Persson et al. [37], in an experimental model of rhinovirus-induced asthma, suggested that IL-1β signaling pathways may be causally involved in the induction of neutrophilic and Th2 characteristics of viral-induced asthma exacerbations.

In our study, the reduction in IL-5 positive cells in the airways was greater in both groups treated with the protease inhibitor (ACO-pep-BbKI and ACO-DX-pep-BbKI) than in the group treated only with a corticosteroid, and there was no difference between the treatments in alveolar septa. Interleukin-5 plays an essential role in the recruitment of eosinophils to the airways, and today, anti-IL-5 and anti-immunoglobulin E (IgE) monoclonal antibodies are already used to reduce the exacerbation of asthma [9,38].

All the treatments were equally efficient in reducing IL-6 positive cells in the airways, but in the alveolar septa, the groups treated with dexamethasone showed an enhanced response compared to the treatment with pep-BbKI alone. Dos Santos et al. [39], in their model of asthma, showed an increase in IL-6 positive cells, a good response to treatment with anti-IL-17, Rho-kinase inhibitor, and the association of both in airways and alveolar septa.

IL-10 and TNF-α positive cells were reduced in the alveolar septa of the group treated only with the pep-BbKI compared to the groups treated with dexamethasone. There are several inflammatory mediators in COPD, including TNF-α that is produced by macrophages and/or respiratory epithelial cells. As it is a potent activator of NF-κB, it may amplify the inflammatory response. In asthma, Th2 cytokines mediate allergic inflammation, and cytokines such as TNF-α and IL-1β amplify the inflammatory response and play a role in more severe diseases [40,41].

In our study, the pep-BbKI group had a similar response in the numbers of IL-13 positive cells relative to the dexamethasone group (ACO-DX). IL-13 mediates allergic responses in patients with asthma and induces bronchial hyperresponsiveness, goblet cell hyperplasia, and mucin production [38]. Although IL-13 being a cytokine involved in asthma, we found high levels of either in the induced emphysema groups (ELA and ACO). Park et al. [42] studied patients with COPD and found high levels of eosinophilia and periostin (a protein involved in airway inflammation), which are produced by the induction of IL-13. These authors showed an improvement in forced expiratory volume in first second (FEV_1_) in these patients when treated with corticosteroids in association with long-acting beta-agonists.

T-helper cells (Th) -17 have been implicated in the processes of several autoimmune and inflammatory diseases, and some authors have suggested that IL-17 may induce IL-1β, IL-6, and TNF-α; furthermore, IL-17 increases the expression of iNOS and is involved in the recruitment of macrophages and neutrophils [43]. Zijlstra et al. [44] suggested that IL-17 is one of the interleukins responsible for inducing resistance of bronchial epithelial cells to the effects of steroids. Our study showed a decrease in counts of IL-17 counts in alveolar septa and airways when treated with pep-BbKI and dexamethasone. Still, in alveolar septa, we found a better response in treatments with protease inhibitor than in the group treated with dexamethasone alone. Some authors have studied treatments with anti-IL-17 in some murine models of elastase-induced injury and asthma, and they found a good response in improving most of the parameters evaluated in inflammation and remodeling [35,39,45].

There are experimental studies in asthma and emphysema models separately, with good results for treatment with serine protease inhibitors. As rBmTI-A reduced the concentrations of IL-5, IL-10, IL-13, and IL-17A, CrataBL reduced the numbers of IFN-γ, IL-4, IL-5, IL-13, and IL-17 positive cells in the airways and alveolar walls; both studies used an asthma model [21,22]. Serine proteases in the emphysema model, such as EcTI and BbKI, showed a reduction in TNF-α levels after treatment with these protease inhibitors [23,24].

Regarding oxidative stress, we showed a reduction in eNO, and a decrease in iNOS-positive cells in all the treated groups, in airways and alveolar walls, with pep-BbKI and with dexamethasone, when compared to the ACO group. Similarly, Martins-Oliveira et al. [23] and Theodoro-Júnior et al. [24] used BbKI and EcTI as treatment in their experimental model of elastase-induced emphysema, and showed an attenuated response to oxidative stress, with a reduction in eNO and decreased expression of iNOS-positive cells in the alveolar septa and airways.

Prado et al. [46], in an experimental asthma model with ovalbumin, showed that eNO and iNOS levels increased in sensitized animals. Exhaled nitric oxide is considered an important marker of airway inflammation in asthma [47], patients with asthma have high levels of eNO, and when they receive treatment with corticosteroids, there is a decline in their levels of eNO [48]. The role of peptidase inhibitors in asthma-COPD overlap experimental models needs to be studied for more comparisons, as they represent a promising therapeutic strategy because corticosteroids have many side effects.

Regarding lung remodeling, there was an increase in the expression of MMP-9 and MMP-12 in the airways and alveolar septa, and for TGF-β only in the alveolar septa, in the overlap group (ACO) compared to the OVA and ELA groups, suggesting a potentiation of the remodeling response in the overlapping disease relative to asthma and emphysema models. Collagen fibers showed an increase in the asthma (OVA) and overlap (ACO) models compared to the control group (SAL), while in the emphysema model (ELA), they did not show an increase. These results likely can be explained because the OVA and ACO groups had more time to be sensitized with ovalbumin and therefore more time to remodeling, and the ELA group received elastase 7 days before the mechanics, with less time to remodeling. This is consistent with studies showing that collagen fibers increase over time after emphysema induction with elastases [49].

In our study, the assessment of extracellular matrix remodeling was based on the evaluation of MMP-9, MMP-12, TGF-β positive cells, and collagen fibers, and showed good results with attenuation of these markers in the airways and alveolar septa in all treatment groups (ACO-pep-BbKI, ACO-DX, ACO-DX-pep-BbKI), compared to the ACO group. Despite the decrease in these markers, some treated groups were still different from the control group (SAL), suggesting that there was attenuation but not a complete reversal of the structural changes. The MMP-12 assessment in alveolar septa showed that the association of the protease inhibitor with dexamethasone (ACO-DX-pep-BbKI) reversed the response relative to ACO and became similar to that in SAL group; furthermore, it showed a more significant decrease in MMP-12 than the other groups (ACO-DX and ACO-pep-BbKI). The collagen fibers differed between treatments only in the evaluation of the alveolar septa; the pep-BbKI group with or without dexamethasone had better attenuation than the ACO-DX group.

Martins-Oliveira et al. [23] showed similar results, with a reduction in MMP-9, MMP-12 positive cells, and volume fraction of collagen fibers in the emphysema group treated with the protease inhibitor BbKI. In addition, Almeida-Reis et al. [20] reported similar results using another protease inhibitor, BbCI, in an elastase-induced model. In an experimental model of asthma, Florencio et al. [22] showed attenuation in MMP-9, TGF-β and collagen fibers after treatment with rBmTI-A, and Bortolozzo et al. [21] using treatment with CrataBL reduced MMP-9, MMP-12 and collagen fibers.

NF-κB is an important indicator of inflammation in lung diseases [50] and has been considered an important regulator of innate and adaptative immune responses. This activation is associated with inflammation, remodeling, and oxidative stress in chronic pulmonary diseases caused by various signals, such as cytokines and pathogens [36,51,52]. In this study, we found an attenuation in NF-κB-positive cells after treatment with pep-BbKI and dexamethasone compared to that in ACO group. In the airways, the ACO-DX and ACO-DX-pep-BbKI groups showed better attenuation than the group treated with pep-BbKI alone, and in alveolar septa, the ACO-DX-pep-BbKI group showed a better response than ACO-pep-BbKI. Bortolozzo et al. [21] obtained similar results in an experimental study using CrataBL to treat chronic allergic pulmonary inflammation.

Even though we have previously study about the effects of BbKI in emphysema mice models, the present study had some limitations, but it is innovative for it fills a gap as too few studies evaluating the use of the protease inhibitor BbKI in the treatment of asthma-COPD overlap in mice.

It is important to note that our study had limitations, including the lack of studies evaluating the use of the protease inhibitor BbKI in treating asthma-COPD overlap in mice. Although we used immunohistochemistry for the analysis, other evaluation methods could be employed to further support our findings. However, we previously demonstrated a good correlation between data obtained by immunohistochemistry and ELISA in models of asthma and emphysema. Despite these limitations, our study had strengths in investigating a wide range of markers to enhance our understanding of asthma-COPD overlap in experimental models. Moreover, we used the BbKI peptide derived from the primary sequence of the kallikrein inhibitor, whereas others utilized the whole protein structure. The pathogenesis of chronic pulmonary diseases such as asthma and COPD is complex and challenging due to the involvement of various pathways and processes. Nevertheless, our results provide a solid foundation for future studies and open up possibilities for exploring new therapeutic approaches for the treatment of asthma-COPD overlap.

## 4. Materials and Methods

### 4.1. Animals

The Ethical Committee approved the study protocol of the University of São Paulo (process number: 1030/2018). Male BALB/c mice aged 6–8 weeks (25–30 g) were acquired from the University of São Paulo and maintained in an animal facility with a 12-h light-dark cycle and were provided water and chow. The animals received humane care in compliance with the “Guide for Care and Use of Laboratory Animals” (National Institutes of Health, publication 86-23, revised 1985). This work was developed in the Laboratory of Experimental Therapy I (LIM-20) of the Faculty of Medicine of the University of São Paulo and was supported by FAPESP (number 2018/02537-5) and CNPq.

### 4.2. Inhibitor Purification

The peptide representing the reactive site sequence of BbKI from *Bauhinia bauhinioides* (P51/62), [53] was purposefully designed and synthesized to identify the minimal structure responsible for its inhibitory function and establish a correlation between the peptide’s structure and the protein’s specificity of action. This synthesized peptide, known as peptide-BbKI (pep-BbKI), bears the sequence RPGLPVRFESPL-NH2. The synthesis of pep-BbKI was carried out by WatsonBio Science, a reputable laboratory located in Texas, USA. The peptide was produced in acetate salts form with a purity level equal to or exceeding 98%, as confirmed by reverse phase chromatography analysis.

### 4.3. Experimental Groups

This protocol was modified by Toledo et al. [54] and Ikeda et al. [27] and lasted for 28 days. Mice were divided at randomly into eight groups (eight animals in each group): (a) control saline SAL (subjected to saline protocol); (b) OVA (ovalbumin, subjected to the OVA sensitization); (c) ELA (subjected to porcine pancreatic elastase); (d) ACO (ovalbumin + porcine pancreatic elastase); (e) ACO-pep-BbKI (subjected to OVA + ELA protocols and treated with BbKI); (f) ACO-DX (subjected to OVA + ELA protocols and treated with dexamethasone); (g) ACO-DX-pep-BbKI (subjected to OVA + ELA protocols and treated with pep-BbKI + dexamethasone); and (h) control saline SAL-pep-BbKI (subjected to saline protocol and treated with pep-BbKI) (Figure 1).

### 4.4. Ovalbumin-Induced Asthma Mice Model

The mice were sensitized by intraperitoneal (IP) injection of 50 mg OVA (A-5378, Sigma Aldrich, St. Louis, MO, USA) emulsified in 6 mg of aluminum hydroxide adjuvant (Pepsamar, Sanofi-Synthelabo SA, Rio de Janeiro, Brazil) on days 1 and 14. The mice were subsequently challenged by OVA inhalation (1% in saline) for 30 min on days 21, 23, 25, and 27 using an ultrasonic nebulizer (US-1000; ICEL, São Paulo, Brazil). Control animals (groups SAL and SAL-BbKI groups) received sterile saline solution instead of OVA (Figure 7). These doses were determined as described in a previous study by Toledo et al. [54].

### 4.5. Elastase-Induced Emphysema Mice Model

For porcine pancreatic elastase (PPE) (E1250-500 mg Elastase from porcine pancreas Type I, ≥4 units/mg protein, 41.7 mL, 12 mg protein/mL; 5 units/mg Protein; Sigma Aldrich, St. Louis, MO, USA) instillation, the mice were anesthetized with inhaled isoflurane (Isofurine^®^ 1 mL/mL, Cristália LTDA, Itapira, SP, Brazil). The experimental animals received intratracheal instillation of 25 U PPE/100 g body weight dissolved in 40 μL of normal saline, on day 21 of the protocol (Figure 7), as previously described by Ikeda et al. [27]. All animals were evaluated after seven days of elastase treatment.

### 4.6. Pep-BbKI and Dexamethasone Treatment Groups

The mice received intraperitoneal injections of 2 mg/kg of the pep-BbKI on days 22, 23, 25, and 27, as previously described by Brito et al. [17]. On these same days of the protocol, the group treated with corticosteroids received intraperitoneal injections of 5 mg/kg of dexamethasone (Aché Laboratórios Farmacêuticos S.A., São Paulo, Brazil), which was described previously by Toledo et al. [54]. The treatments were administered one hour before each ovalbumin nebulization, when asthma was induced in the experimental group according to protocol (Figure 7).

### 4.7. Evaluation of Hyperresponsiveness to Methacholine and Determination of Exhaled Nitric Oxide

On day 28 of the protocol, 24 h after the last inhalation with ovalbumin or saline, mice were anesthetized using thiopental (50 mg/kg IP) and tracheostomized, and lung mechanics were measured using a computer-controlled small animal ventilator (FlexiVent, Scireq, Montreal, QC, Canada). The animals were ventilated at 150 breaths/min with a tidal volume of 10 mL/kg. Data recording began when the animals were motionless. For the determination of exhaled nitric oxide (eNO) an impermeable balloon (Mylar Bag, Sievers, Instruments Inc., Boulder, CO, USA) was connected to the expiratory portion of the ventilator for 10 min, after which the balloons were connected to another machine for identification of the eNO results, which were measured by chemiluminescence using a rapid response analyzer (280 NOA-Nitric Oxide Analyzer; Sievers Instruments Inc., Boulder, CO, USA). After the collection of the eNO balloon, the animals were subjected to an evaluation of the respiratory system mechanics. Pressure values were obtained, and the impedance of the airway (pressure/flow) was calculated as a function frequency. Using a pop-up signal of 75% in 16 s, 3 blocks of 8 s were used to calculate the parameters of the mechanical oscillation (*i*), according to the following equation:𝑍(𝑓) = Raw + 𝑖(2𝜋𝑓)𝑙aw + [Gtis − 𝑖Htis]/(2𝜋𝑓)^𝛼^

In this model, 𝑍(𝑓) is the impedance of air as a function of frequency, 𝑖 is the imaginary unit (−1.5), 𝑓 is frequency, 𝑙aw is the inertance of the airways, and 𝛼 = (2/𝜋) ∗ arctan (Htis/Gtis). The parameters experimentally obtained were airway resistance (Raw), tissue resistance (Gtis), and tissue elastance (Htis) by using a constant phase model that can separately analyze airways and tissue. Resistance of the respiratory system (Rrs) and elastance (Ers) were also analyzed according to the equation of motion:Paw = *FlowR* + *VolE* + *P*0

In this equation Paw (cmH_2_O) is the airway pressure, *Flow* (mL/s) is the airway flow, *R* (cmH_2_O.s/mL) is the resistance of respiratory system, and *Vol* (mL) is the air volume of the lung since the beginning of inspiration, *E* (cmH_2_O/mL) is elastance of respiratory system and *P*0 (cmH_2_O) is the airway pressure in the inspiration beginning [55,56]. The dose-response curve to methacholine was performed, containing 3 doses of the drug (3, 30 and 300 mg/mL) in the first, second and third minute, in order to evaluate the bronchoconstrictor response of the airways and lung parenchyma; the results of such measures were expressed as a percentage increase from baseline for the variables captured through the mechanical ventilator. After these procedures, mice were exsanguinated and a bronchoalveolar lavage fluid analysis was performed. Afterward, the lungs were removed and fixed at constant pressure, 20 cmH_2_O, for 24 h in 4% formaldehyde.

### 4.8. Bronchoalveolar Lavage Fluid (BALF) Analysis

BALF was obtained by introducing saline (0.5 mL) into the lungs via a tracheal cannula three times (volume 1.5 mL) and withdrawing the fluid into a test tube. The collected fluid was centrifuged at 1000 rpm, for 10 min, at 5 °C, and the cell pellet was resuspended in 200 μL of sterile saline. Total cells were counted using a Neubauer chamber (400×). For the differential cell counts, 100 μL of the BAL was cytocentrifuged at 450 rpm for 6 min and performed using a cytospin slide preparation stained with Diff-Quick Reagent. Differential cell counts were performed by evaluating 300 cells using an optical microscope and using the morphological criteria [32].

### 4.9. Morphometric Studies

The lungs were cut, and after fixation, the material was subjected to the usual histological techniques with paraffin to obtain 4 µm thick slices and the slides were stained with hematoxylin-eosin. The quantification of mean linear intercept (Lm) was performed using an optical microscope with an integrated eyepiece containing a known area (10^4^ μm^2^ at a magnification of 1000×) of 50 lines and 100 points. We analyzed 10 randomly selected fields for each animal.

To analyze collagen fibers, the slides were stained with Picrosirius red for 1 h at room temperature and then washed in running water for 5 min. After this step, the slides were stained with Harris hematoxylin for 6 min and then washed in running water for 10 min. We measured the area of collagen in the total area of the alveolar septa and around the airway, which was 10–12 microscopic fields per lung. The calculation was performed through the measurements using an optical microscope with the aid of an image analyzer (Image ProPlus 4.5 for Windows 98/NT/2000), and the optical density measurement was used to detect the collagen fibers. Images were captured at a magnification of 400× using a Leica DFC 420 digital camera (Leica, Wetzlar, Germany) coupled to a DM2500 optical microscope (Leica). The images were sent to another computer where they were processed using the Qwim Plus program (Leica) and analyses were performed using the Image ProPlus software 4.5 (NIH). The collagen content was expressed as a percentage, as the relationship between the quantity of collagen in a specific frame and the total area of that frame (volume fraction).

Slides were also prepared for immunohistochemical staining to evaluate the cellular expression of interleukin (IL)-1β, IL-4, IL-5, IL-6, IL-10, IL-13, IL-17, Interferon (IFN-γ), tumor necrosis factor (TNF-α), metalloproteinase (MMP) 9 and MMP-12, transforming growth factor (TGF-β), inducible nitric oxide synthase (iNOS), and nuclear factor kappa B (NF-κB).

Morphometric analysis was performed using an optical microscope, and we used a reticulum with 50 lines and 100 points, using the point counting technique [57]. The reticulum was positioned in the lung parenchyma, and we analyzed 10 fields of alveolar walls. For airways the reticulum was positioned adjacent to the airway walls, and 12 fields were analyzed per animal. The number of points in the positive cells was counted on the number of points in the alveolar septa and airway wall for each field chosen. We evaluated them at 1000 magnification. The results were expressed as the percentages (%) of the positive area (volume fraction).

### 4.10. Immunohistochemistry

Immunohistochemistry was performed according to the method described by Camargo et al. [35]. The antibodies used for the markers and dilutions in this study are listed in Table 3. Digestion of the antigen at high temperature was performed in a pressure cooker for 1 min using citrate buffer (6.0 pH). For incubation with the primary antibody, the sections were diluted in bovine serum albumin (BSA) solution and applied to each slide. Then, the slides were incubated overnight in a humid chamber at 4 °C for 18–22 h. The slides were then washed in phosphate-buffered saline (PBS) and incubated with the secondary antibody using the ABC Kit with Vectastain (Vector Laboratories, CA, USA) (Table 3). To visualize the positive cells, the slides were washed in PBS and proteins were visualized using diaminobenzidine (DAB) chromogen (70 mg DAB in 110 mL of the Tris-HCl; Sigma Chemical Co., St. Louis, MO, USA) and Harris hematoxylin (Merck, Darmstadt, Germany), and finally mounted with microscopy resin.

### 4.11. Data Analysis

Statistical analysis was performed using the Scientific Graphing Software SigmaPlot^®^ Version 11.0. To determine the difference between groups and their statistical significance, we used the unidirectional analysis of variance One Way ANOVA followed by the *Holm–Sidak* method for multiple comparisons and *t*-test for two comparisons. All data are represented as display graphs as bar plots, which indicate the mean and standard error. Statistical significance was set at *p* < 0.05.

## 5. Conclusions

We concluded that the experimental model of asthma-COPD overlap was effective. Compared to the control group saline, the animals showed exacerbation in the parameters of airway hyperresponsiveness in the challenges with methacholine, worsening oxidative stress, remodeling parameters, and an increase in inflammation markers. The peptide BbKI and dexamethasone were equally effective in reducing inflammation, remodeling, and oxidative stress in an experimental model of ACO. Therefore, a protease inhibitor such as BbKI requires more studies to facilitate comparisons and become a potential therapeutic strategy for asthma-COPD overlap.

## Figures and Tables

**Figure 1 ijms-24-11261-f001:**
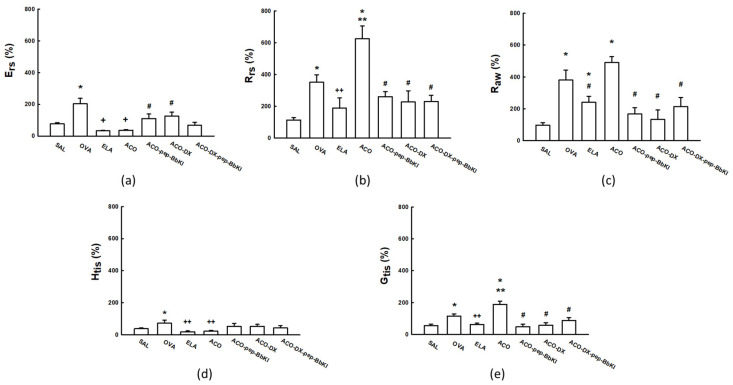
Mechanical evaluation of (**a**) Ers, respiratory system elastance; (**b**) Rrs, respiratory system resistance; (**c**) Raw, airway resistance; (**d**) Htis, lung tissue elastance; and (**e**) Gtis, tissue resistance. The results are expressed in percentages (%). * *p* < 0.05 compared to SAL group; ** *p* < 0.05 compared to OVA and ELA groups; # *p* < 0.05 compared to ACO group; + *p* < 0.05 compared to SAL and OVA groups; ++ *p* < 0.05 compared to OVA group. N = 8 for each group.

**Figure 2 ijms-24-11261-f002:**
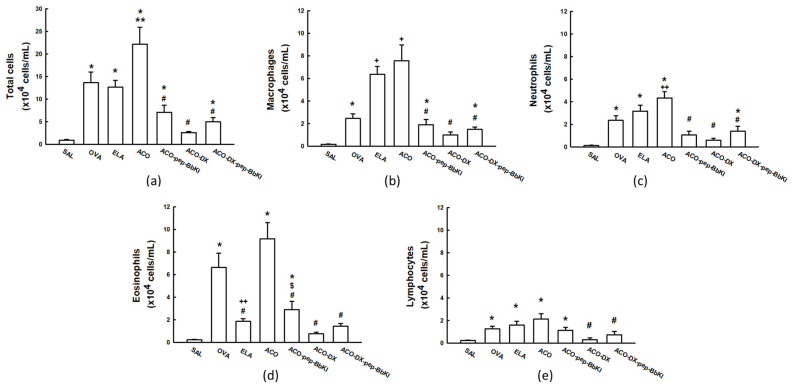
Evaluation of the number of cells in bronchoalveolar lavage fluid (BALF). The (**a**) total numbers of cells, (**b**) macrophages, (**c**) neutrophils, (**d**) eosinophils, and (**e**) lymphocytes are expressed in ×10^4^ cells/mL. * *p* < 0.05 compared to SAL group; ** *p* < 0.05 compared to OVA and ELA groups; # *p* < 0.05 compared to ACO group; + *p* < 0.05 compared to SAL and OVA groups; ++ *p* < 0.05 compared to OVA group; $ *p* < 0.05 compared to ACO-DX group. N = 8 for each group.

**Figure 3 ijms-24-11261-f003:**
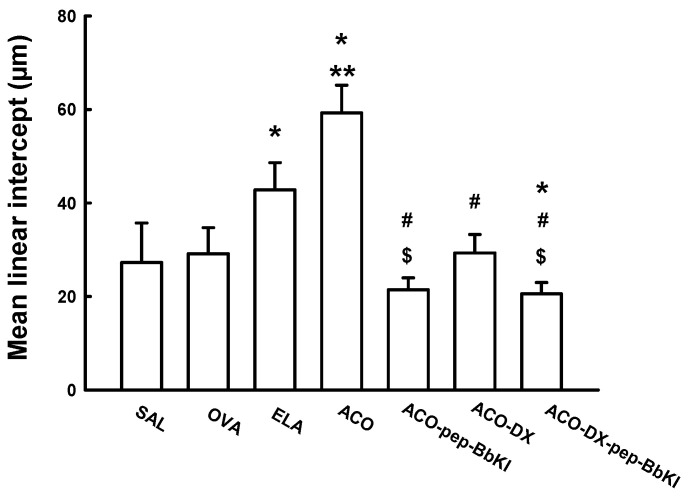
Evaluation of mean linear intercepts; the results are expressed in micrometers (μm). * *p* < 0.05 compared to SAL group; ** *p* < 0.05 compared to OVA and ELA groups; # *p* < 0.05 compared to ACO group; $ *p* < 0.05 compared to ACO-DX group. N = 8 for each group.

**Figure 4 ijms-24-11261-f004:**
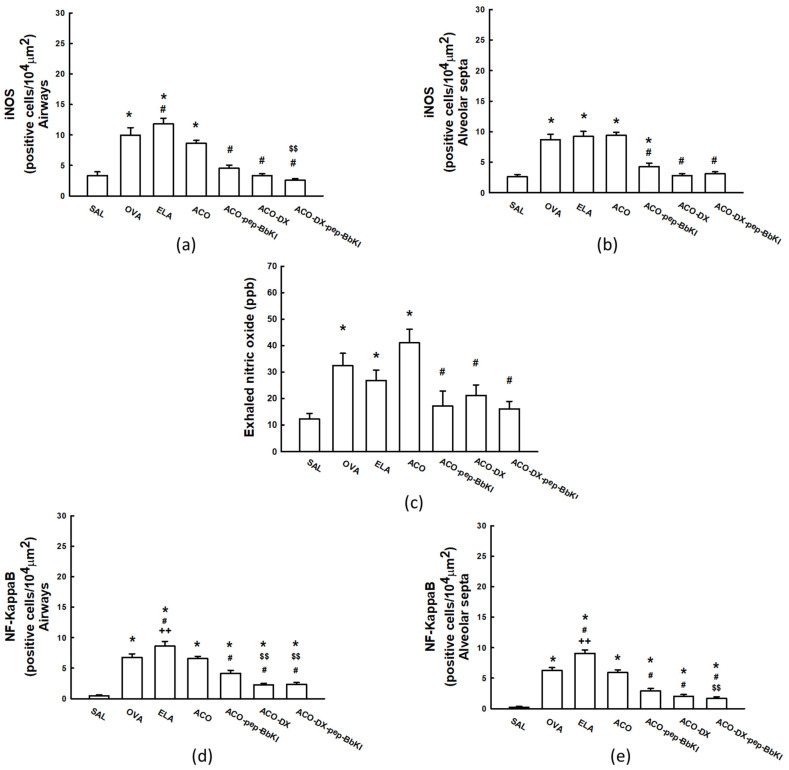
Graphs of evaluation of oxidative stress and signaling pathway: (**a**) iNOS in airways and (**b**) iNOS in alveolar septa are expressed in cells/10^4^ µm^2^; (**c**) exhaled nitric oxide is expressed in ppb; (**d**) NF-κB in airways and (**e**) NF-κB in alveolar septa are expressed in cells/10^4^ µm^2^. * *p* < 0.05 compared to SAL group; # *p* < 0.05 compared to ACO group; ++ *p* < 0.05 compared to OVA group; $$ *p* < 0.05 compared to ACO-pep-BbKI group. N = 8 to each group. iNOS, inducible nitric oxide synthase; NF, nuclear factor.

**Figure 5 ijms-24-11261-f005:**
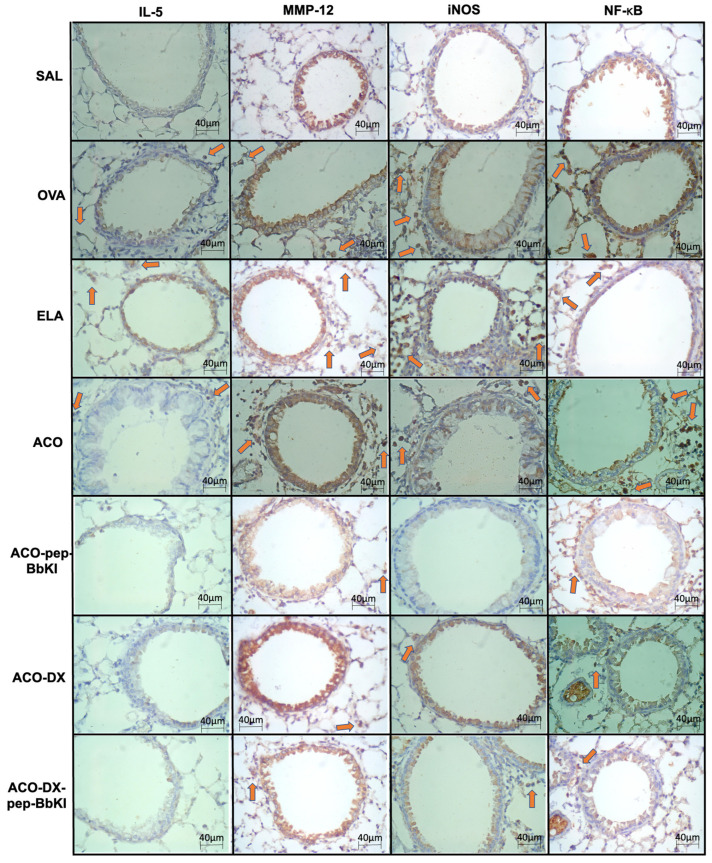
Qualitative analysis of inflammatory marker (IL-5), remodeling marker (MMP-12), oxidative stress (iNOS), and signaling pathway (NF-κB). Photomicrographs of the results of the immunohistochemical analyses show the presence of inflammation around the airways. The red arrows indicate positive cells for IL-5, MMP-12, iNOS and NF-κB. Magnification of 400×. The experimental groups include SAL, OVA, ELA, ACO, ACO-pep-BbKI, ACO-DX, and ACO-DX-pep-BbKI. IL, interleukin; MMP, metalloproteinase; iNOS, inducible nitric oxide synthase; NF, nuclear factor.

**Figure 6 ijms-24-11261-f006:**
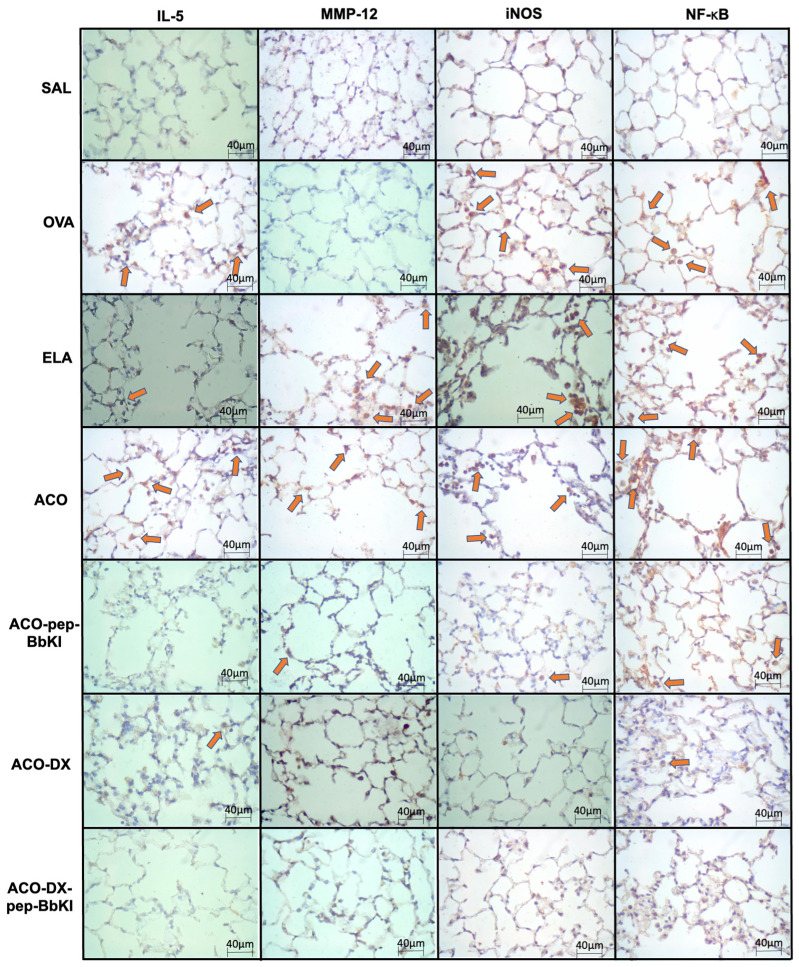
Qualitative analysis of inflammatory marker (IL-5), remodeling marker (MMP-12), oxidative stress (iNOS), and signaling pathway (NF-κB). Photomicrographs of the results of the immunohistochemical analyses show the presence of inflammation in the alveolar septa. The red arrows indicate positive cells for IL-5, MMP-12, iNOS and NF-κB. Magnification of 400×. The experimental groups include SAL, OVA, ELA, ACO, ACO-pep-BbKI, ACO-DX, and ACO-DX-pep-BbKI. IL, interleukin; MMP, metalloproteinase; iNOS, inducible nitric oxide synthase; NF, nuclear factor.

**Figure 7 ijms-24-11261-f007:**
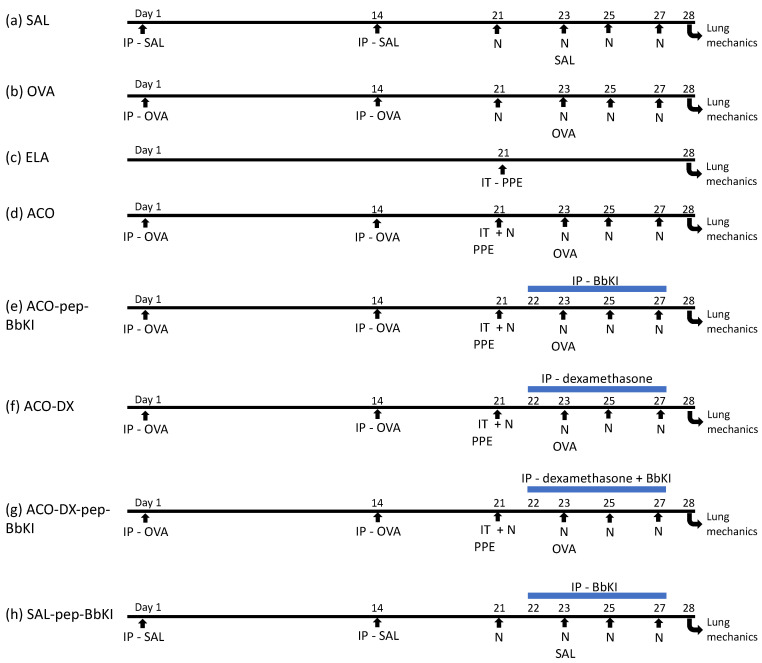
Schematic of the experimental protocols. (**a**) Control mice received saline intraperitoneal (Days 1 and 14) and nebulization with saline (Days 21, 23, 25, and 27); (**b**) Mice were sensitized with ovalbumin intraperitoneally (Days 1 and 14) and received nebulization with ovalbumin (Days 21, 23, 25 and 27); (**c**) Mice received intratracheal elastase (Day 21); (**d**) Mice received intraperitoneal ovalbumin (Days 1 and 14), intratracheal elastase (Day 21) and nebulization with ovalbumin (Days 21, 23, 25 and 27); (**e**) Mice received intraperitoneal ovalbumin (Days 1 and 14), intratracheal elastase (Day 21), nebulization with ovalbumin (Days 21, 23, 25 and 27) and were treated with intraperitoneal pep-BbKI (Days 22, 23, 25 and 27); (**f**) Mice received intraperitoneal ovalbumin (Days 1 and 14), intratracheal elastase (Day 21) and nebulization with ovalbumin (Days 21, 23, 25 and 27), and were treated with intraperitoneal dexamethasone (Days 22, 23, 25 and 27); (**g**) Mice received intraperitoneal ovalbumin (Days 1 and 14), intratracheal elastase (Day 21), nebulization with ovalbumin (Days 21, 23, 25 and 27) and were treated with intraperitoneal dexamethasone and pep-BbKI (Days 22, 23, 25 and 27); (**h**) Mice received intraperitoneal saline (Days 1 and 14) and nebulization with saline (Days 21, 23, 25, and 27) and were treated with pep-BbKI (Days 22, 23, 25 and 27). IP, intraperitoneal injection; IT, intratracheal instillation; N, nebulization; PPE, porcine pancreatic elastase.

**Table 1 ijms-24-11261-t001:** Absolute values of the inflammatory markers.

Inflammatory Markers (Cells/10^4^ µm^2^)	SAL	OVA	ELA	ACO	ACO-pep-BbKI	ACO-DX	ACO-DX-pep-BbKI
IL-1β—Airways	0.4 ± 0.1	4.2 ± 0.4 *	1.2 ± 0.2 ^++^	6.0 ± 0.5 *^/^**	2.7 ± 0.3 *^/#/$^	1.4 ± 0.3 *^/#^	2.2 ± 0.2 *^/#^
IL-1β—Alveolar septa	0.2 ± 0.1	2.6 ± 0.3 *	0.7 ± 0.2 ^++^	4.0 ± 0.3 *^/^**	0.9 ± 0.2 *^/#^	0.5 ± 0.1 ^#^	1.3 ± 0.2 *^/#^
IL-4—Airways	1.5 ± 0.5	11.3 ± 1.3 *	4.1 ± 1.2 ^#/++^	12.2 ± 2.3 *	11.9 ± 1.7 *	7.9 ± 1.2 *	8.7 ± 1.2 *
IL-4—Alveolar septa	1.7 ± 0.3	10.4 ± 0.6 *	4.7 ± 0.7 *^/++^	7.9 ± 0.7 */**	8.8 ± 0.7 *	6.5 ± 0.5 *	6.5 ± 0.7 *
IL-5—Airways	1.7 ± 0.2	11.3 ± 0.7 *	6.5 ± 0.5 *^/++^	8.5 ± 0.6 */**	2.4 ± 0.4 ^#/$^	6.6 ± 0.5 *^/#^	2.7 ± 0.5 ^#/$^
IL-5—Alveolar septa	1.2 ± 0.2	4.4 ± 0.3 *	2.9 ± 0.6 *^/++^	8.0 ± 0.4 */**	1.9 ± 0.3 ^#^	2.0 ± 0.2 ^#^	1.7 ± 0.2 ^#^
IL-6—Airways	0.7 ± 0.4	8.3 ± 0.5 *	5.5 ± 0.4 *^/#/++^	7.4 ± 0.6 *	2.2 ± 0.2 *^/#^	1.9 ± 0.3 *^/#^	1.8 ± 0.2 *^/#^
IL-6—Alveolar septa	0.4 ± 0.1	6.6 ± 0.4 *	7.2 ± 0.5 *	12.3 ± 1.1 *^/^**	1.9 ± 0.2 *^/#^	1.1 ± 0.2 ^#/$$^	1.0 ± 0.2 ^#/$$^
IL-10—Airways	2.1 ± 0.2	3.3 ± 0.2 *	4.1 ± 0.3 *	5.2 ± 0.4 *^/++^	2.8 ± 0.3^#/$^	4.1 ± 0.3 *	2.3 ± 0.2 ^#/$^
IL-10—Alveolar septa	3.2 ± 0.3	4.5 ± 0.3 *	5.5 ± 0.4 *	6.6 ± 0.4 *^/++^	2.3 ± 0.2^#^	3.6 ± 0.3 ^#/$$^	3.3 ± 0.3 ^#/$$^
IL-13—Airways	1.7 ± 0.2	10.4 ± 0.7 *	7.9 ± 0.7 *^/#/++^	11.6 ± 0.6 *	4.7 ± 0.4 *^/#^	5.8 ± 0.4 *^/#^	4.3 ± 0.3 *^/#/$^
IL-13—Alveolar septa	2.6 ± 0.4	9.2 ± 0.6 *	6.7 ± 0.6 *^/++^	16.9 ± 1.5 *^/^**	5.2 ± 0.4 *^/#^	4.2 ± 0.3 ^#^	6.0 ± 0.9 *^/#^
IL-17—Airways	2.1 ± 0.2	5.6 ± 0.3 *	6.4 ± 0.3 *	7.3 ± 0.4 ^+^	3.1 ± 0.2 ^#^	4.0 ± 0.3 *^/#^	3.7 ± 0.2 *^/#^
IL-17—Alveolar septa	1.5 ± 0.2	6.5 ± 0.3 *	8.1 ± 0.4^+^	8.4 ± 0.6 ^+^	2.7 ± 0.2 ^#/$^	4.9 ± 0.3 *^/#^	3.2 ± 0.6 *^/#/$^
IFN-γ—Airways	0.7 ± 0.2	2.1 ± 0.2	3.3 ± 0.4 *	6.2 ± 0.7 *^/^**	1.1 ± 0.2 ^#/$^	0.3 ± 0.1 ^#^	1.9 ± 0.3 *^/#/$^
IFN-γ—Alveolar septa	0.7 ± 0.2	2.9 ± 0.3 *	2.4 ± 0.3 *	3.2 ± 0.3 *	1.2 ± 0.2 ^#^	0.6 ± 0.2 ^#^	0.9 ± 0.2 ^#^
TNF-α—Airways	1.7 ± 0.4	6.2 ± 0.7 *	5.3 ± 0.7 *	5.0 ± 0.7 *	2.8 ± 0.4	5.2 ± 0.7 *^/$$^	4.9 ± 0.7 *^/$$^
TNF-α-Alveolar septa	1.3 ± 0.2	5.5 ± 0.4 *^/#^	6.2 ± 0.6 *	7.7 ± 0.5 *	2.1 ± 0.3 ^#^	4.5 ± 0.4 *^/#/$$^	4.8 ± 0.4 *^/#/$$^

Notes: IL, interleukin; IFN-γ, interferon-gamma; TNF, tumor necrosis factor. The results are expressed in positive cells/10^4^ µm^2^. * *p* < 0.05 compared to SAL group; ** *p* < 0.05 compared to OVA and ELA groups; # *p* < 0.05 compared to ACO group; + *p* < 0.05 compared to SAL and OVA groups; ++ *p* < 0.05 compared OVA group; $ *p* < 0.05 compared to ACO-DX group; $$ *p* < 0.05 compared to ACO-pep-BbKI group. N = 8 for each group.

**Table 2 ijms-24-11261-t002:** Absolute values of remodeling markers.

Remodeling Markers	SAL	OVA	ELA	ACO	ACO-pep-BbKI	ACO-DX	ACO-DX-pep-BbKI
MMP-9—Airways (cells/10^4^ µm^2^)	0.2 ± 0.03	3.0 ± 0.1 *	5.3 ± 0.2 *	10.2 ± 0.3 */**	1.8 ± 0.08 *^/#^	2.1 ± 0.08 *^/#^	1.7 ± 0.1 *^/#/$^
MMP-9—Alveolar septa (cells/10^4^ µm^2^)	0.5 ± 0.1	2.1 ± 0.08 *	4.4 ± 0.1 *	8.7 ± 0.3 *^/^**	1.9 ± 0.1 *^/#^	2.3 ± 0.1 *^/#^	1.5 ± 0.1 *^/#/$^
MMP-12—Airways (cells/10^4^ µm^2^)	1.3 ± 0.2	1.9 ± 0.2	2.1 ± 0.4	3.5 ± 0.4 *^/^**	1.8 ± 0.2 ^#^	0.9 ± 0.1 ^#/$$^	1.0 ± 0.2 ^#/$$^
MMP-12—Alveolar septa (cells/10^4^ µm^2^)	0.4 ± 0.1	0.2 ± 0.08	2.7 ± 0.4 *	5.7 ± 0.5 *^/^**	2.5 ± 0.3 *^/#^	2.2 ± 0.3 *^/#^	0.9 ± 0.2 ^#/$/$$^
TGF-β—Airways (cells/10^4^ µm^2^)	0.8 ± 0.3	5.5 ± 1.1 *	2.7 ± 0.4 ^#^	7.0 ± 0.9 *	3.9 ± 0.4 *^/#^	3.0 ± 0.4 *^/#^	1.6 ± 0.2 ^#/$$^
TGF-β—Alveolar septa (cells/10^4^ µm^2^)	0.1 ± 0.07	4.8 ± 0.3 *	3.6 ± 0.4 *	6.4 ± 0.5 *^/^**	2.3 ± 0.3 *^/#/$^	0.7 ± 0.2 ^#^	1.9 ± 0.2 *^/#/$^
Collagen fibers—Airways (%)	1.6 ± 0.4	12.5 ± 0.4 *	0.3 ± 0.1 ^++/#^	12.0 ± 1.7 *	1.8 ± 0.4 ^#^	1.3 ± 0.2 ^#^	3.0 ± 2.4 ^#^
Collagen fibers—Alveolar septa (%)	2.4 ± 0.2	11.3 ± 0.3 *	3.9 ± 0.5 ^++^	9.0 ± 0.8 *^/^**	2.9 ± 0.3 ^#/$^	5.7 ± 0.7 *^/#^	2.5 ± 0.2 ^#/$^

Note: MMP, metalloproteinase; TGF, transforming growth factor. The MMPs and TGF-β are expressed in positive cells/10^4^ µm^2^. The collagen fibers are expressed in percentages (%). * *p* < 0.05 compared to SAL group; ** *p* < 0.05 compared to OVA and ELA groups; # *p* < 0.05 compared to ACO group; ++ *p* < 0.05 compared to OVA group; $ *p* < 0.05 compared to ACO-DX group; $$ *p* < 0.05 compared to ACO-pep-BbKI group.

**Table 3 ijms-24-11261-t003:** Markers, specifications and dilutions.

Marker	Specifications Primary Antibody	Dilution	Secondary Antibody	Specifications Secondary Antibody
IL-1β	SC-52012, L: A0719; Santa Cruz Biotechnology, CA, USA.	1:50	anti-mouse	L: ZG0715, Vector; Vectastain Elite ABC Kit Peroxidase (Mouse IgG), CA, USA.
IL-4	SC-53084, L: J1518; Santa Cruz Biotechnology, CA, USA.	1:8000	anti-mouse	L: ZF0206, Vector; Vectastain Elite ABC Kit Peroxidase (Mouse IgG), CA, USA.
IL-5	SC-398334, L: F1617; Santa Cruz Biotechnology, CA, USA.	1:300	anti-mouse	L: ZF0206, Vector; Vectastain Elite ABC Kit Peroxidase (Mouse IgG), CA, USA.
IL-6	LS-C746886, L: 144178; LSBio, WA, USA.	1:200	anti-rabbit	L: ZF0103, Vector; Vectastin Elite ABC Kit (Rabbit IgG), CA, USA.
IL-10	SC-8438; Santa Cruz Biotechnology, CA, USA.	1:50	anti-mouse	L: ZF0206, Vector; Vectastain Elite ABC Kit Peroxidase (Mouse IgG), CA, USA.
IL-13	SC-393365, L:G1715; Santa Cruz Biotechnology, CA, USA.	1:8000	anti-mouse	L: ZF0206, Vector; Vectastain Elite ABC Kit Peroxidase (Mouse IgG), CA, USA.
IL-17	SC-7927, L:A3113; Santa Cruz Biotechnology, CA, USA.	1:100	anti-rabbit	L: ZF0103, Vector; Vectastin Elite ABC Kit (Rabbit IgG), CA, USA.
IFN-γ	SC-8308, L:B2811; Santa Cruz Biotechnology, CA, USA.	1:100	anti-rabbit	L: ZF0103, Vector; Vectastin Elite ABC Kit (Rabbit IgG), CA, USA.
TNF-α	SC-52746, L: J2418; Santa Cruz Biotechnology, CA, USA.	1:5000	anti-mouse	L: ZF0206, Vector; Vectastain Elite ABC Kit Peroxidase (Mouse IgG), CA, USA.
MMP-9	SC-393859, L: 6118; Santa Cruz Biotechnology, CA, USA.	1:800	anti-mouse	L: ZF0206, Vector; Vectastain Elite ABC Kit Peroxidase (Mouse IgG), CA, USA.
MMP-12	SC-30072, L: B1910; Santa Cruz Biotechnology, CA, USA.	1:400	anti-rabbit	L: ZF0103, Vector; Vectastin Elite ABC Kit (Rabbit IgG), CA, USA.
TGF-β	SC-130348, L: A0219; Santa Cruz Biotechnology, CA, USA.	1:700	anti-mouse	L: ZF0206, Vector; Vectastain Elite ABC Kit Peroxidase (Mouse IgG), CA, USA.
iNOS	RB-9242-P, L: 9242P709C; Thermo Fisher Scientific, UK.	1:150	anti-rabbit	L: ZF0103, Vector; Vectastin Elite ABC Kit (Rabbit IgG), CA, USA.
NF-κB	SC-8008, L: B1119; Santa Cruz Biotechnology, CA, USA.	1:700	anti-mouse	L: ZF0206, Vector; Vectastain Elite ABC Kit Peroxidase (Mouse IgG), CA, USA.

Note: IL, interleukin; IFN, interferon; TNF, tumor necrosis factor; MMP, metalloproteinase; TGF, transforming growth factor; iNOS, inducible nitric oxide synthase; NF, nuclear factor; CA, California; USA, United States of America; UK, United Kingdom.

## Data Availability

Data are contained within the article and Appendix A.

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
