# Peer review of "Effects of a Peptide Derived from the Primary Sequence of a Kallikrein Inhibitor Isolated from *Bauhinia bauhinioides* (pep-BbKI) in an Asthma–COPD Overlap (ACO) Model"

_ijms, 2023, doi:10.3390/ijms241411261_

Round 1
Reviewer 1 Report
The manuscript in its present form is too long and very complicated. The authors made a lot of different experiments. In my opinion, you should consider splitting the description of the presented results into two independent manyscripts. With such a long discussion, the reader forgets what the purpose of the experiment was.
The style needs minor adjustments.
Author Response
Effects of a peptide derived from the primary sequence of kallikrein inhibitor isolated from Bauhinia bauhinioides (pep-BbKI) in an asthma-COPD overlap (ACO) model - Luana Laura Sales da Silva et al.
International Journal of Molecular Sciences
The 2022-2023 Journal's Impact IF of International Journal of Molecular Sciences is 6.208, which is just updated in 2023.
ijms-2455003 - Reply to Reviewers
Dear Editor of IJMS and ijms@mdpi.com
We would like to thank the reviewers for their thoughtful guidance and considerations. Through them we were able to improve the manuscript and demonstrate its objectives.
We will answer point by point all the comments. We did a marked draft with the changes in evidence with the different color and underline. The text removed was marked with simple strikethrough. We also performed a clean copy with the revised manuscript, and we expect that these corrections clarify and improve the manuscript.
Reviewer 1
Comments and Suggestions for Authors
Point 1
The manuscript in its present form is too long and very complicated. The authors made a lot of different experiments. In my opinion, you should consider splitting the description of the presented results into two independent manuscripts. With such a long discussion, the reader forgets what the purpose of the experiment was.
Response 1
We modified the discussion as requested.
Point 2
Comments on the Quality of English Language
The style needs minor adjustments.
Response 2
We thank the reviewer for this consideration.

Reviewer 2 Report
The latest research paper by Luana Laura Sales da Silva deals with a difficult clinical problem, the asthma-COPD overlap and the use of a natural occurring peptide from Bauhinia bauhinioides in its treatment. This is a beautiful research with a detailed experimental part, well written and presented. The results are well discussed and results of other research groups are presented in details showing that protease inhibitors from plants might be of interest in treatment of asthma or emphysema.
Nevertheless, I would have some remarks and questions to the authors.
Was the aim the aim of the present study the influence of BbKI on the progression of the disease or rather on the control of symptoms (see line 100)? Both attempts are fine but the first one would require some analyzes during the OVA sensitization (before and after treatment).
In line 453, what did you mean by “Mouse models of emphysema or asthma are more common than experimental models for asthma and emphysema overlap”? It seems to be quite obvious that a model of a certain disease is used when its symptoms are needed even if the disease is more complex. I do agree that the model you decided to use is a very interesting and in fact, good model of this overlap.
Consider to add more details about the use of protease inhibitors in respiratory diseases (line 459).
Could you discuss a bit the statement “BbKI (as a protein structure, not a peptide)”? It is not clear.
Line 563: all interleukins are cytokines, so maybe it would be enough to write about cytokines in general.
The sentence in line 686 “Our study had some limitations, including the lack of studies evaluating the use of protease inhibitor BbKI in treating asthma-COPD overlap in mice” suggests that you did not study the asthma/copd overlap, which in fact, you did. Consider to reformulate it.
Check everywhere if the units (esp. in Tables) are correct.
Author Response
Effects of a peptide derived from the primary sequence of kallikrein inhibitor isolated from Bauhinia bauhinioides (pep-BbKI) in an asthma-COPD overlap (ACO) model - Luana Laura Sales da Silva et al.
International Journal of Molecular Sciences
The 2022-2023 Journal's Impact IF of International Journal of Molecular Sciences is 6.208, which is just updated in 2023.
ijms-2455003 - Reply to Reviewers
Dear Editor of IJMS and ijms@mdpi.com
We would like to thank the reviewers for their thoughtful guidance and considerations. Through them we were able to improve the manuscript and demonstrate its objectives.
We will answer point by point all the comments. We did a marked draft with the changes in evidence with the different color and underline. The text removed was marked with simple strikethrough. We also performed a clean copy with the revised manuscript, and we expect that these corrections clarify and improve the manuscript.
For the convenience of the revisors we retyped the questions.
Reviewer 2
Comments and Suggestions for Authors
The latest research paper by Luana Laura Sales da Silva deals with a difficult clinical problem, the asthma-COPD overlap and the use of a natural occurring peptide from Bauhinia bauhinioides in its treatment. This is a beautiful research with a detailed experimental part, well written and presented. The results are well discussed and results of other research groups are presented in details showing that protease inhibitors from plants might be of interest in treatment of asthma or emphysema.
Nevertheless, I would have some remarks and questions to the authors.
Point 1
Was the aim the aim of the present study the influence of BbKI on the progression of the disease or rather on the control of symptoms (see line 100)? Both attempts are fine but the first one would require some analyzes during the OVA sensitization (before and after treatment).
Response 1
As we worked with an experimental model, we evaluated the control of functional histopathological alterations of inflammation and remodeling, which we consider having an impact on the progression of the disease.
Point 2
In line 453, what did you mean by “Mouse models of emphysema or asthma are more common than experimental models for asthma and emphysema overlap”? It seems to be quite obvious that a model of a certain disease is used when its symptoms are needed even if the disease is more complex. I do agree that the model you decided to use is a very interesting and in fact, good model of this overlap.
Response 2
We completely agree with the referee. In line 453, we meant to say that “Mouse models of emphysema and asthma overlap are quite unusual because they are less studied until the moment”. In order to make it clear, we changed the phrase in the text, as:
“New therapies have been studied using mouse models of emphysema or asthma, but experimental models for asthma and emphysema overlap still lacking in the literature.”
Point 3
Consider to add more details about the use of protease inhibitors in respiratory diseases (line 459).
Response 3
Thanks for your consideration. In fact, we have detailed the use of protease inhibitors from the line 477 throughout all the discussion, accordingly to the studied features.
Point 4
Could you discuss a bit the statement “BbKI (as a protein structure, not a peptide)”? It is not clear.
Response 4
Thanks for your appointment. In the study of Martins-Oliveira et al., 2016, it was used the protein as protease inhibitor. In our study the peptide that presents the sequence of the reactive site of BbKI, from Bauhinia bauhinioides, was designed and synthesized based on the BbKI protein sequence to establish the smallest structure responsible for the inhibitory function and to correlate with the structure and specificity of action of the protein.
In order to make it clear, we changed the text as:
“(it was used as protein, and not the peptide)”.
Point 5
Line 563: all interleukins are cytokines, so maybe it would be enough to write about cytokines in general.
Response 5
Line 563: All interleukins are cytokines: great observation, thank you! We rephrased the sentence "These ILs and cytokines" to "These cytokines…”.
Point 6
The sentence in line 686 “Our study had some limitations, including the lack of studies evaluating the use of protease inhibitor BbKI in treating asthma-COPD overlap in mice” suggests that you did not study the asthma/copd overlap, which in fact, you did. Consider to reformulate it.
Response 6
Line 686 “Our study had some limitations, including the lack of studies evaluating the use of protease inhibitor BbKI in treating asthma-COPD overlap in mice”. We agree with the referee and reformulated the sentence as:
“It is important to note that our study had limitations, including the lack of studies evaluating the use of the protease inhibitor BbKI in treating asthma-COPD overlap in mice.”
Point 7
Check everywhere if the units (esp. in Tables) are correct.
Response 7
According to the reviewer's suggestion, we checked all units (especially in Tables) and throughout the text.

Reviewer 3 Report
The authors look at the effect of a peptide derived from the primary sequence of the kallikrein inhibitor isolated from Bauhinia bauhinioides (pep-BbKI) in an asthma-COPD overlap (ACO) model. This is interesting animal model work but has some gaps:
- The abstract has to be 200 words and without paragraphs.
- It lacks an abbreviations section.
- In keywords they put asthma-chronic obstructive pulmonary disease overlap syndrome. It is no longer considered a syndrome.
- Lines 97-98: Therefore, different protease inhibitors of plant origin have been studied in asthma and COPD models. The references of these studies are missing.
- In table 1 because they only put the SAL group and d SAL-pep-BbKI. To see that there is no difference with treatment? This could go in supplementary material.
- In the figures modify the x-axis, it is not necessary to put the same in all the figures. Improve their quality.
- Why have you not quantified cytokines in BAL?
- NF-κB is localized in the nucleus?
- Abbreviations should be put in all figure captions.
- If they already had control of asthma and COPD why haven't they taken advantage and looked at the effect of pep-BbKI in these pathologies?
Author Response
Effects of a peptide derived from the primary sequence of kallikrein inhibitor isolated from Bauhinia bauhinioides (pep-BbKI) in an asthma-COPD overlap (ACO) model - Luana Laura Sales da Silva et al.
International Journal of Molecular Sciences
The 2022-2023 Journal's Impact IF of International Journal of Molecular Sciences is 6.208, which is just updated in 2023.
ijms-2455003 - Reply to Reviewers
Dear Editor of IJMS and ijms@mdpi.com
We would like to thank the reviewers for their thoughtful guidance and considerations. Through them we were able to improve the manuscript and demonstrate its objectives.
We will answer point by point all the comments. We did a marked draft with the changes in evidence with the different color and underline. The text removed was marked with simple strikethrough. We also performed a clean copy with the revised manuscript, and we expect that these corrections clarify and improve the manuscript.
For the convenience of the revisors we retyped the questions.
Reviewer 3
Comments and Suggestions for Authors
The authors look at the effect of a peptide derived from the primary sequence of the kallikrein inhibitor isolated from Bauhinia bauhinioides (pep-BbKI) in an asthma-COPD overlap (ACO) model. This is interesting animal model work but has some gaps:
Point 1
- The abstract has to be 200 words and without paragraphs.
Response 1
Thanks for your carefully revision. We have adapted and concatenated part of the abstract to follow the correct specifications of the abstract in general, that must be no longer than 200 words. The Abstract should serve both as a general introduction to the topic and as a brief, non-technical summary of the main results and their implications.
Point 2
- It lacks an abbreviations section.
Response 2
Thanks for the observation. We included an abbreviations section.
Point 3
- In keywords they put asthma-chronic obstructive pulmonary disease overlap syndrome. It is no longer considered a syndrome.
Response 3
The reviewer is dead right: in keywords we put asthma-chronic obstructive pulmonary disease overlap syndrome, where ACO is no longer considered a syndrome. We fixed that. Nevertheless we would like to point out that we were not able to find an adequate descriptor at the NIH - MeSH (Medical Subject Headings).
Point 4
- Lines 97-98: Therefore, different protease inhibitors of plant origin have been studied in asthma and COPD models. The references of these studies are missing.
Response 4
Thanks for the comments. Accordingly to the request of the reviewer, we have included references in the text, such as the following:
- Almeida-Reis, R.; Theodoro-Junior, O.A.; Oliveira, B.T.M.; Oliva, L.V.; Toledo-Arruda, A.C.; Bonturi, C.R.; Brito, M.V.; Lopes, F.D.T.Q.S.; Prado, C.M.; Florencio, A.C.; Martins, M.A.; Owen, C.A.; Leick, E.A.; Oliva, M.L.V.; Tibério, I.F.L.C. Plant Proteinase Inhibitor BbCI Modulates Lung Inflammatory Responses and Mechanic and Remodeling Alterations Induced by Elastase in Mice. Biomed Res Int. 2017, 2017, 8287125. doi: 10.1155/2017/8287125.
- Bortolozzo, A.S.S.; Rodrigues, A.P.D.; Arantes-Costa, F.M.; Saraiva-Romanholo, B.M.; de Souza, F.C.R.; Brüggemann, T.R.; de Brito, M.V.; Ferreira, R.D.S.; Correia, M.T.D.S.; Paiva, P.M.G.; Prado, C.M.; Leick, E.A.; Oliva, M.L.V.; Martins, M.A.; Ruiz-Schutz, V.C.; Righetti, R.F.; Tibério, I.F.L.C. The Plant Proteinase Inhibitor CrataBLPlays a Role in Controlling Asthma Response in Mice. Biomed Res Int. 2018, 2018, 9274817. doi: 10.1155/2018/9274817.
- Florencio, A.C.; de Almeida, R.S.; Arantes-Costa, F.M.; Saraiva-Romanholo, B.M.; Duran, A.F.; Sasaki, S.D.; Martins, M.A.; Lopes, F.D.T.Q.S.; Tibério, I.F.L.C.; Leick, E.A. Effects of the serine protease inhibitor rBmTI-A in an experimental mouse model of chronic allergic pulmonary inflammation. Sci Rep.2019, 9, 12624. doi: 10.1038/s41598-019-48577-4. Erratum in: Sci Rep. 2019, 9, 17011.
- Martins-Olivera, B.T.; Almeida-Reis, R.; Theodoro-Júnior, O.A.; Oliva, L.V.; Neto Dos Santos Nunes, N.; Olivo, C.R.; Vilela de Brito, M.; Prado, C.M.; Leick, E.A.; Martins, Mde.A.; Oliva, M.L.; Righetti, R.F.; Tibério, Ide.F. The Plant-Derived Bauhinia bauhinioides Kallikrein Proteinase Inhibitor (rBbKI) Attenuates Elastase-Induced Emphysema in Mice. Mediators Inflamm. 2016, 2016, 5346574. doi: 10.1155/2016/5346574.
- Theodoro-Júnior, O.A.; Righetti, R.F.; Almeida-Reis, R.; Martins-Oliveira, B.T.; Oliva, L.V.; Prado, C.M.; Saraiva-Romanholo, B.M.; Leick, E.A.; Pinheiro, N.M.; Lobo, Y.A.; Martins, M.A.; Oliva, M.L.; Tibério, I.F. A Plant Proteinase Inhibitor from Enterolobium contortisiliquum Attenuates Pulmonary Mechanics, Inflammation and Remodeling Induced by Elastase in Mice. Int J Mol Sci. 2017, 18, 403. doi: 10.3390/ijms18020403.
- Oliva, L.V.; Almeida-Reis, R.; Theodoro-Junior, O.; Oliveira, B.M.; Leick, E.A.; Prado, C.M.; Brito, M.V.; dos Santos Correia, M.T.; Paiva, P.M.G.; Martins, M.A.; Oliva, M.L.V.; Tibério, I.F.L.C. A plant proteinase inhibitor from Crataeva tapia (CrataBL) attenuates elastase-induced pulmonary inflammatory, remodeling, and mechanical alterations in mice. Process Biochemistry 2015, 50, 1958-1965. doi: https://doi.org/10.1016/j.procbio.2015.06.004.
- Rodrigues, A.P.D.; Bortolozzo, A.S.S.; Arantes-Costa, F.M.; Saraiva-Romanholo, B.M.; de Souza, F.C.R.; Brüggemann, T.R.; Santana, F.P.R.; de Brito, M.V.; Bonturi, C.R.; Nunes, N.N.D.S.; Prado, C.M.; Leick, E.A.; Oliva, M.L.V.; Martins, M.A.; Righetti, R.F.; Tibério, I.F.L.C. A plant proteinase inhibitor from Enterolobium contortisiliquum attenuates airway hyperresponsiveness, inflammation and remodeling in a mouse model of asthma. Histol Histopathol. 2019, 34, 537-552. doi: 10.14670/HH-18-059.
Point 5
- In table 1 because they only put the SAL group and d SAL-pep-BbKI. To see that there is no difference with treatment? This could go in supplementary material.
Response 5
Your suggestion is very pertinent. We put this in supplementary material.
Point 6
- In the figures modify the x-axis, it is not necessary to put the same in all the figures. Improve their quality.
Response 6
Your suggestion is valid. Nonetheless, we did not have enough time to perform the changes. We apologize for that.
Point 7
- Why have you not quantified cytokines in BAL?
Response 7
We did not quantify cytokines in BAL in this study, because the evaluation of the lung tissue can increase the pathophysiological understanding, localizing the cellular expression in the airways and alveolar septa. Furthermore, we have previously demonstrated this assessment in the BAL and confirmed the results by PCR:
Camargo, L.D.N.; Righetti, R.F.; Aristóteles, L.R.C.R.B.; Dos Santos, T.M.; de Souza, F.C.R.; Fukuzaki, S.; Cruz, M.M.; Alonso-Vale, M.I.C.; Saraiva-Romanholo, B.M.; Prado, C.M.; Martins, M.A.; Leick, E.A.; Tibério, I.F.L.C. Effects of Anti-IL-17 on Inflammation, Remodeling, and Oxidative Stress in an Experimental Model of Asthma Exacerbated by LPS. Front Immunol. 2018, 8, 1835. doi: 10.3389/fimmu.2017.01835.
Point 8
- NF-κB is localized in the nucleus?
Response 8
In resting cells, most NF-κB is bound to the IκB inhibitor protein, which maintains the complex in the cytoplasm. After appropriate stimulation, the IκB protein is phosphorylated and degraded, leading to the transcription of NFkB in the cell nucleus. The antibody used in this study marks the p65 unit in the cell's cytoplasm.
Point 9
- Abbreviations should be put in all figure captions.
Response 9
As for abbreviations, at the suggestion of the reviewer, we have included in the figures legends.
Point 10
- If they already had control of asthma and COPD why haven't they taken advantage and looked at the effect of pep-BbKI in these pathologies?
Response 10
Previously, we showed the effects of BbKI in models of emphysema. In order to seek future therapeutic implications, we developed the BbKI peptide. Although the effect was likely to be maintained, we tested this peptide in a model of asthma and COPD and improved our understanding of the effect of this peptide using an ACO model. As shown, this model had an increment in all evaluated parameters, compared to the asthma and COPD model. Furthermore, the peptide demonstrated a better effect than the protein itself.

Reviewer 4 Report
The results and discussion are OK to be acceptable with the reason why this research's priority. However, I would recommend the authors to revise their writing of materials and methods. The points are informed in the followings.
1. BALB/c mice were bred and maintained in SPF or conventional conditions ?
2. pep-BbKI purification is simply described, but more intensive explanation should be described for the readers.
3. Did BALB/c mice in each group (N=8) survived all until the end of the experimental period ? In results, authors should notice the number of each group of mice for analysis in legends for evaluating statistical analyses.
Author Response
Effects of a peptide derived from the primary sequence of kallikrein inhibitor isolated from Bauhinia bauhinioides (pep-BbKI) in an asthma-COPD overlap (ACO) model - Luana Laura Sales da Silva et al.
International Journal of Molecular Sciences
The 2022-2023 Journal's Impact IF of International Journal of Molecular Sciences is 6.208, which is just updated in 2023.
ijms-2455003 - Reply to Reviewers
Dear Editor of IJMS and ijms@mdpi.com
We would like to thank the reviewers for their thoughtful guidance and considerations. Through them we were able to improve the manuscript and demonstrate its objectives.
We will answer point by point all the comments. We did a marked draft with the changes in evidence with the different color and underline. The text removed was marked with simple strikethrough. We also performed a clean copy with the revised manuscript, and we expect that these corrections clarify and improve the manuscript.
For the convenience of the revisors we retyped the questions.
Reviewer 4
Comments and Suggestions for Authors
The results and discussion are OK to be acceptable with the reason why this research's priority. However, I would recommend the authors to revise their writing of materials and methods. The points are informed in the followings.
Point 1
BALB/c mice were bred and maintained in SPF or conventional conditions?
Response 1
Thanks for your question. BALB/c mice were SPF but after the beginning of the experiments were bred and maintained under conventional conditions.
Point 2
pep-BbKI purification is simply described, but more intensive explanation should be described for the readers.
Response 2
We modified the methodology in the text to better clarify the peptide:
The peptide representing the reactive site sequence of BbKI from Bauhinia bauhinioides (P51/62), was purposefully designed and synthesized to identify the minimal structure responsible for its inhibitory function and establish a correlation between the peptide's structure and the protein's specificity of action. This synthesized peptide, known as peptide-BbKI (pep-BbKI), bears the sequence RPGLPVRFESPL-NH2. The synthesis of pep-BbKI was carried out by WatsonBio Science, a reputable laboratory located in Texas, USA. The peptide was produced in acetate salts form with a purity level equal to or exceeding 98%, as confirmed by reverse phase chromatography analysis.
Sumikawa, J.T.; Brito, M.V.; Macedo, M.L.; Uchoa, A.F.; Miranda, A.; Araujo, A.P.; Silva-Lucca, R.A.; Sampaio, M.U.; Oliva, M.L. The defensive functions of plant inhibitors are not restricted to insect enzyme inhibition. Phytochemistry. 2010, 7, 214-20. doi: 10.1016/j.phytochem.2009.10.009.
Point 3
Did BALB/c mice in each group (N=8) survived all until the end of the experimental period ? In results, authors should notice the number of each group of mice for analysis in legends for evaluating statistical analyses.
Response 3
We agree with the reviewer that this is very important to be explained. In our experiment, BALB/c mice in each group (N=8)
